# Vascular plants affect properties and decomposition of moss-dominated peat, particularly at elevated temperatures

Lilli Zeh[1], Marie Theresa Igel[1], Judith Schellekens[2], Juul Limpens[3], Luca Bragazza[4], Karsten Kalbitz[1]

[1] Soil Resources and Land Use, Institute of Soil Science and Site Ecology, Technische Universität Dresden, Pienner Str. 19, 01737 Tharandt, Germany

[2] Departamento de Ciência do Solo, Escola Superior de Agricultura ''Luiz de Queiroz'' – ESALQ/USP, Av. Pádua Dias 11, Piracicaba, São Paulo, Brazil

[3] Plant Ecology and Nature Conservation, Wageningen University, Droevendaalse steeg 3a, 6708 PB, Wageningen, The Netherlands

[4] Agroscope, Plant Production Systems, Route de Duillier 50, P.O. Box 1012, 1260 Nyon, Switzerland

*Correspondence to*: Lilli Zeh (lilli.zeh@tu-dresden.de)

**Abstract.** Peatlands, storing significant amounts of carbon are extremely vulnerable to climate change. The effects of climate change are projected to lead to a vegetation shift from *Sphagnum* mosses to sedges and shrubs. Impacts on the present moss-dominated peat remain largely unknown. In this study, we used a multi proxy approach to investigate the influence of contrasting vascular plant types (sedges, shrubs) on peat chemistry and decomposition. Peat cores of 20 cm depth and plant material (*Sphagnum spp., Calluna vulgaris, Eriophorum vaginatum*) from two ombrotrophic peatlands in the Italian Alps with a mean annual temperature difference of 1.4 °C were analysed. Peat cores were taken under adjacent shrub and sedge plants growing at the same height above the water table. We used carbon, nitrogen and their stable isotopes to assess general patterns in the degree of decomposition across sampling locations and depths. In addition, analytical pyrolysis was applied to disentangle effects of vascular plants (sedge, shrub) on chemical properties and decomposition of the moss-dominated peat. Pyrolysis data confirmed that *Sphagnum* moss dominated the present peat irrespectively of depth. Nevertheless, vascular plants contributed to peat properties as revealed by e.g. pyrolysis products of lignin. The degree of peat decomposition increased with depth as shown by e.g. decreasing amounts of the pyrolysis product of sphagnum acid and increasing $\delta^{13}C$ with depth. Multiple parameters also revealed a higher degree of decomposition of *Sphagnum*-dominated peat collected under sedges than under shrubs, particularly at the high temperature site. Surprisingly, temperature effects on peat decomposition were less pronounced than those of sedges. Our results imply that vascular plants affect the decomposition of the existing peat formed by *Sphagnum*,

particularly at elevated temperature. These results suggest that changes in plant functional types may have a stronger impact on the soil carbon feedback in a warmer world than hitherto assumed.

## 1 Introduction

Peatland soils store about 550 Gt of carbon (C), which equals one third of all soil organic C while they only cover 3 % of the world's land area (Parish et al., 2008). In contrast to mineral soils, C sequestration in peatlands is not controlled by stabilisation processes related to soil minerals (Schmidt et al., 2011), but is environmentally constrained by low temperatures and prevalent anoxic conditions (high water tables). Climate change is expected to partly lift these environmental constraints on microbial decomposition by warmer (Karhu et al., 2014) and drier conditions, threatening to release stored organic C as $CO_2$ to the atmosphere. Alterations in the environment will also initiate shifts in vegetation composition, generally favouring vascular plants (sedge, shrub) over *Sphagnum* (moss) (Berendse et al., 2001; Breeuwer et al., 2009; Heijmans et al., 2008; Malmer et al., 1994). A systematic change in composition of plant functional types (PFTs) towards vascular plants has a yet unknown potential to accelerate C losses from the stored peat originally dominated by mosses due to increased C input via roots from vascular plants (Bragazza et al., 2013; Gavazov et al., 2018; Robroek et al., 2015) and litter mixing effects (Zhang et al., 2019).

Vascular plants in alpine peatlands were shown to have up to twice as high net biomass production as mosses (Gerdol et al., 2010) and consequently relatively higher litter inputs than mosses. In addition to litter quantity, the chemical composition differs between PFTs with considerable consequences on decomposition dynamics. *Sphagnum* litter tends to decompose slower than vascular plant litter (Bragazza, 2006; Coulson and Butterfield, 1978; Verhoeven and Toth, 1995; Zhang et al., 2019) due to high carbon to nitrogen (C/N) ratio and decay-inhibiting structural carbohydrates (Coulson and Butterfield, 1978; Hájek et al., 2011; Schellekens et al., 2015b; Turetsky et al., 2008). Within vascular plants, shrub litter differentiates from sedge litter by higher C/N ratio and lower decay rate (Huang et al., 1998; Kaštovská et al., 2018; Laiho et al., 2003; Limpens and Berendse, 2003). Analog to biomass production, shrubs and sedges have higher belowground input of fresh root litter (Schellekens et al., 2011) and C input of living roots in comparison to mosses (i.e. fresh photosynthates; Zeh et al., 2019). Growth and reproduction of microorganisms are supposed to increase with higher root activity (Bragazza et al., 2015; Ward et al., 2013) which is likely to stimulate peat decomposition. Translating the PFT properties outlined above into consequences for C storage of

autochthonous peats where inputs of all species are mixed together remains a challenge, particularly if the impact of recently changing vegetation on the previously formed peat is of interest. This challenge calls for a multi-proxy approach (Biester et al., 2014) for determining the impact of the varying properties of PFTs on peat decomposition *in situ*.

Various analytical approaches exist to assess peat properties and decomposition, each with its own advantages and drawbacks. Carbon to N ratios have been widely used to evaluate decomposition of peat (Biester et al., 2014; Broder et al., 2012; Kuhry and Vitt, 1996; Limpens and Berendse, 2003; Taylor et al., 1989). The microbial mineralisation of C-rich compounds and the subsequent respiration and emission as $CO_2$ decreases the abundance of C relative to N (Broder et al., 2012). In combination with a high immobilisation of N by microbial biomass, N will be further enriched in the remaining organic material (Damman,

1988). Consequently, C/N ratios are decreasing with increasing decomposition. However, PFTs influence the C/N ratio of peat, too, as their litter differ in C and N contents (Hornibrook et al., 2000).

The stable isotopes $^{13}C$ and $^{15}N$ and their vertical trends have been often used alongside C/N ratios to reflect changes in decomposition (Biester et al., 2014; Broder et al., 2012; Coolen and Orsi, 2015; Krüger et al., 2014, 2015; Novák et al., 1999). Aerobic peat decomposition leads to an enrichment in $\delta^{13}C$ and $\delta^{15}N$, due to a preferential use of the lighter isotopes by

microorganisms and hence a relative enrichment of $^{13}C$ and $^{15}N$ in the remaining organic matter (Alewell et al., 2011; Bragazza et al., 2010; Bragazza and Iacumin, 2009; Kalbitz et al., 2000; Nadelhoffer and Fry, 1988). However, stable isotope patterns are also affected by the water table limiting aerobic decomposition and thus isotopic discrimination in the remaining peat (Krüger et al., 2015) and the plant species forming the litter. Sedge leaves were found to be more enriched in $^{13}C$ and $^{15}N$ than shrub leaves. The isotopic ratios for living plant parts found in this study are consistent with the ranges reported in previous

studies. Sedge leaves were found to vary in $\delta^{13}C$ signature between -27.0 to -26.85 ‰ and in $\delta^{15}N$ between -3.96 to -0.9 ‰; reported ranges for $\delta^{13}C$ in shrub leaves are -29.2 to -28.83 ‰ and for $\delta^{15}N$ -10.92 to -9.7 ‰ (Biester et al., 2014; Gavazov et al., 2016; Ménot and Burns, 2001; Nordbakken et al., 2003). *Sphagnum* samples were found to vary in $\delta^{13}C$ signature between -30.4 and -25.0 ‰ (Bragazza and Iacumin, 2009; Loisel et al., 2009; Preis et al., 2018; Price et al., 1997; Proctor et al., 1992) and in $\delta^{15}N$ signatures between -7.5 and 2.5 ‰ (Asada et al., 2005; Biester et al., 2014; Bragazza et al., 2005; Kohzu et al.,

2003; Ménot and Burns, 2001; Nordbakken et al., 2003).

Pyrolysis gas chromatography/mass spectrometry (py-GC/MS) is a powerful but labour intensive tool to characterise the composition of peat, and disentangle the effects of source material from decomposition (Abbott et al., 2013; Huang et al., 1998; McClymont et al., 2011; Schellekens et al., 2012, 2015a, 2015b). Plant specific pyrolysis products which have been used to distinguish vascular plants from *Sphagnum* in peat include lignin-phenols from lignin and 4-isopropenylphenol from

sphagnum acid, respectively (Van Der Heijden et al., 1997; McClymont et al., 2011; Schellekens et al., 2009, 2015b, 2015c).

In this multi-proxy study, we combined the analytical approaches outlined above to explore the influence of vascular plants on chemical properties and degree of peat decomposition in two moss-dominated peatlands contrasting in temperature. We hypothesized that: i) chemical properties of the moss-dominated peat differ under shrub and sedge coverage, (ii) the decomposition of the moss-dominated peat increases with depth and is higher under sedge than shrub coverage, and iii)

increasing temperature is reflected in higher degree of decomposition of the moss-dominated peat, particularly under sedge coverage.

## 2 Material and Methods

### 2.1 Study sites

Two ombrotrophic peatlands at different altitude in the north-eastern Alps of Italy were chosen to simulate a climate warming

scenario. The peatland at lower altitude, Lupicino, is located at 1290 m a.s.l. and is characterised by a mean annual temperature (MAT) of 6.3°C and a total annual precipitation of 810 mm. The peatland at higher altitude, Palù Tremole, is located at 1700 m a.s.l. with a MAT of 4.9°C and total annual precipitation of 825 mm. Soil temperature in 10 cm depth between August 2015 and July 2016 was 7.1°C at Lupicino and 5.9°C at Palù Tremole. Lupicino will be referred to as High T site and Palù Tremole as Low T site. The time during which the top 20 cm of the peat was above the water table was determined with water table

measurements between August 2015 and July 2016 at three gauges on each site. At the Low T site, the water table remained below 20 cm for 137, 138 and 284 days of the year; at the High T site, this was 117, 360 and 366 days of the year respectively.. Furthermore, in peatlands with similar vegetation cover, situated at 1030 m a.s.l. and 1880 m a.s.l. in Switzerland, the age of peat in 15-20 depth was found to be 40 years and 26 year, respectively (Gavazov et al., 2018), meaning that the potential age difference between the peat sampled at our sites is likely less than 15 years. Vegetation community on both sites is similar and

dominated by *Sphagnum spp.*, with contributions from *Calluna vulgaris* and *Eriophorum vaginatum*, representing the three

PFTs (bryophyte mosses, ericoid shrubs, graminoid sedges). This experimental setup offers a unique opportunity to disentangle

impacts of shrubs and sedges on properties and decomposition of a moss-dominated peat. Further detail in biotic and climatic

conditions can be found in Zeh et al. (2019).

## 2.2 Sampling and preparation

During the first half of August 2015, we selected 20 hummocks per study site with a closed peat moss cover of at least 95 %,

an equal proportion of shrub to sedge cover and a total vascular plant cover of $47 \pm 2$ % at Low T site and $77 \pm 2$ % at the High

T site. The hummocks were located in five groups (blocks) of four hummocks, with maximally 5 m between the hummocks

within a block. On each hummock, we took two peat cores: one directly under *C. vulgaris* (shrub-core) and one under *E.*

*vaginatum* tussocks (sedge-core), yielding a total of 40 cores per peatland. Peat was defined as all organic bulk material

accumulating underneath the peatlands surface, comprised of a matrix of mostly dead *Sphagnum* material with embedded

living and dead stems and roots of *C. vulgaris* and *E. vaginatum*. Peat cores were sampled with a custom-made metal peat

corer with an inside diameter of 5 cm and sampling length of 20 cm. Additionally, photosynthetically active moss tissues from

the top 2 cm and plant shoots from *E. vaginatum* and *C. vulgaris* were collected from each hummock. Furthermore, six separate

peat cores with 10 cm diameter (three at each site) were randomly sampled for collection of living roots of shrubs and sedges.

During the field campaign, peat cores were stored at 8°C within PVC tubes to prevent deformation. Afterwards, they were

frozen to -20°C until further sample preparation.

Of the collected 80 peat cores, the 20 most representative cores were selected using three criteria: i) peat cores with a minimum

length of 20 cm and without physical damages, ii) one shrub-core and one sedge-core from each block and peatland, and iii)

peat cores with the smallest deviation from the mean weight of the respective block. From the active moss tissues, samples

were selected corresponding to the chosen peat cores and mixed, if not sampled from the same hummock.

To assess which depth increments are appropriate to characterise changes in peat properties, four cores (one sedge and one

shrub from each site) were randomly chosen from the selected 20. These cores were cut into depth increments of 1 cm, except

for the topmost increment, which accounted for 2 cm. Carbon and N concentrations were measured as described in Sect. 2.3.

The results of these analyses indicated changes in C and N concentrations at 2 cm, 5 cm and 12 cm. Considering these results,

the remaining 16 peat cores were cut into four depth increments: 0-2 cm, 2-5 cm, 5-12 cm, 12-20 cm.

Plant and peat material was freeze-dried and then grinded (Fritsch pulverisette 23) before being analysed.

### 2.3 Total carbon and nitrogen concentration and stable isotope analysis ($^{13}C$, $^{15}N$)

Carbon and N concentration were measured with a Vario El III elemental analyser (Elementar Analysensysteme GmbH, Langenselbold, Germany), following standard processing techniques. Carbon and N concentrations were calculated based on

total sample weight. The C/N ratio represents the atomic relationship between C and N content of the peat material.

Isotope analysis was done with vario PYRO CUBE coupled to the visION IRMS (Isoprime, Elementar Analysensysteme GmbH, Langenselbold, Germany). Stable C isotope ratios are reported as $\delta^{13}C$ in [‰] relative to the V-PDB standard and stable N isotope ratios as $\delta^{15}N$ in [‰] relative to air.

### 2.4 Pyrolysis gas chromatography/mass spectrometry

To identify chemical properties of PFTs, peat and peat decomposition, representative plant samples for each PFT and one shrub-core and one sedge-core from each peatland were selected to be additionally analysed by py-GC/MS. We defined peat decomposition as any changes in properties of the bulk peat relative to its source material, e.g. plant material from *C. vulgaris*, *E. vaginatum*, *Sphagnum spp.* directly after deposition.

Plant samples comprised of one root and one shoot sample of shrub and sedge respectively, and one moss tissue sample from

each peatland. In total ten plant samples and four cores were cut into four increments as previously described. They were chosen based on the lowest deviation from the mean in C/N, $\delta^{13}C$ and $\delta^{15}N$ analyses. A Pyrolyser EGA/PY-3030D device (Frontier Laboratories, Fukushima, Japan) has been used for analysis. The pyrolysis temperature was set at 600°C, held for 10 s. The pyrolyser was connected with a GC 7890B and MS 5977 (Agilent Technologies, St. Clara, United States). Inlet temperature of the GC was 250°C (split 50:1). The GC oven had an initial temperature of 45°C (held for 4 min), was than

heated to 240°C at 4°C min$^{-1}$ and afterwards heated to 300°C at 39°C min$^{-1}$ (held for 15 min). The GC column, a ZB-5ms (Zebron, Phenomenex Inc., Torrance, United States), had a length of 30 m, a film thickness 0.25 µm, and a diameter of 0.25 mm. The MS was scanning in the range of 50-600 *m/z*.

Pyrolysis product identification and peak integration in pyrograms were performed with Masslab. Based on intensity and frequency on the total ion current (TIC), 57 pyrolysis products were selected for quantification in all 26 samples (A1).

Quantification was based on the peak area of characteristic fragment ions ($m/z$) for each product (A1). The relative proportion of each product was expressed as a percentage of the total quantified peak area in one sample (TIC: 100 %). The products were grouped according to chemical similarity and their source into: $n$-alkenes and $n$-alkanes, lignin-derived products, phenols, benzenes and polysaccharides.

## 2.5 Selection of molecular parameters

Based on the results of previous pyrolysis studies from peatlands a number of pyrolytic parameters reflecting plant species and the degree of peat decomposition were extracted (Table 1). A pyrolysis product specific for sphagnum acid (4-isopropenylphenol; Van Der Heijden et al., 1997) has been found to very sensitively reflect aerobic decomposition of *Sphagnum* tissue in *Sphagnum*-dominated peat (Schellekens et al., 2015b). Methoxyphenols are unique to lignin, thereby providing a measure for the contribution from vascular plants in peat dominated by *Sphagnum*, because *Sphagnum* contains

no lignin (Abbott et al., 2013; Kracht and Gleixner, 2000; Schellekens et al., 2015c; van Smeerdijk and Boon, 1987). Since both shrubs and sedges contain lignin, additional parameters were included to distinguish between them. Sedges have large contributions from p-coumaric and ferulic acid (Lu and Ralph, 1999) with typical pyrolysis products 4-vinylphenol (Lg1) and 4-vinylguaiacol (Lg4), respectively (Van Der Hage et al., 1993). Because 4-vinylphenol is also abundant in *Sphagnum* tissue (van Smeerdijk and Boon, 1987), the ratio of 4-vinylguaiacol to the summed guaiacyl products (G) can therefore be used to

reflect sedges (Schellekens et al., 2012). The ratio of $C_3$-guaiacol to G usually reflects intact lignin in soils but has been found indicative for shrubs in peat (Schellekens et al., 2012, 2015a). $n$-Alkenes and $n$-alkanes (Al) originate from cutan and suberan present in roots and bark (Nierop, 1998; Tegelaar et al., 1995) and leaf waxes (Eglinton and Hamilton, 1967), depending on their chain length, all of which are associated with shrubs in *Sphagnum*-dominated peat (Schellekens and Buurman, 2011; van Smeerdijk and Boon, 1987).

## 2.6 Statistics

All data analysis and visualisation were performed with R, Version 3.6 (R Core Team, 2019). The preliminary four peat cores which were cut into 1 cm increments were integrated in statistical analysis. For this purpose, the means of C [%], N [%], C/N ratio, $\delta^{13}C$ [‰] and $\delta^{15}N$ [‰] were calculated for each depth increment (2-5 cm, 5-12 cm and 12-20 cm). Linear mixed-effects models (LMM) were applied to results of C/N ratio, $\delta^{13}C$ and $\delta^{15}N$ analysis to consider the nested structure of peat cores in blocks using packages "lme4" (Bates et al., 2015). Vascular plant effect (shrub vs. sedge), site (Low T vs. High T) and depth increments were analysed as fixed effect factors, whereby the peat cores nested in blocks were accounted for as random factor with random intercept structure. If the assumptions of normality in the data and their residuals were met (checked with Shapiro-Wilk test, histograms and Q-Q plots), a subsequent analysis of variance type II with a Kenward-Roger approximation of degree of freedom was applied on the linear-mixed models for hypothesis testing using package "lmerTest" (Kuznetsova et al., 2017). Post-hoc test was accomplished with a pairwise Tukey test applying package "emmeans" (Lenth, 2019). Since data and residuals of C/N ratio, $\delta^{13}C$ and $\delta^{15}N$ in shoot materials were not normally distributed, single effects were tested with Wilcoxon test. Due to limited root sample size, hypothesis tests were neglected.

Principal component analysis (PCA) was applied to the py-GC/MS data. The aim was to reduce the data to a set of uncorrelated, meaningful components. Each principal component is determined by the largest variance (or largest remaining variance) of all quantified pyrolysis products and therefore explains a specific percentage of the total variance. They may represent a single effect on peat chemistry, while the abundance of individual pyrolysis products (or other variables) may be influenced by several environmental factors. Scores indicate to which extend each principal component contributes to a sample. Loadings demonstrate which individual pyrolysis products are responsible for the patterns in scores. Prior to PCA, the number of pyrolysis products (variables) was adapted. First, the *n*-alkenes and *n*-alkanes were pooled to a single variable since these products were large in number (10 and 9, respectively) but in sum only contributed less than 1 % to total abundance. Accordingly, it was avoided that this large set of correlated variables with small abundance was allocated to the first principal component (Schellekens et al., 2017). Second, key molecular parameters (Sect. 2.5, Table 1) were included as variables to check their interpretation within the studied plant samples and peat cores. This resulted in a total number of 41 variables for

PCA. To unravel the effects of vascular plants and temperature on peat decomposition in detail, regression analysis and depth

records of these molecular parameters are discussed (Sect. 4.1.2, 4.2).

## 3 Results

### 3.1 Carbon to nitrogen ratios and isotopic composition of PFTs and peat

Carbon to N ratios and isotopic composition of plant shoots were different between plant tissues, PFTs and to a lesser extent

between sites (Fig. 1). Living plant shoots had consistently lower C/N ratios than plant roots irrespective of site or PFT (Fig.

1a). Sedge shoots had significantly lower C/N ratios ($p < 0.001$, Fig. 1a) and were significantly enriched in $^{13}C$ ($p < 0.001$, Fig.

1b) compared to moss and shrub shoots. Shrub shoots were significantly depleted in $\delta^{15}N$ compared to sedge and moss shoots

($p < 0.001$, Fig. 1c). Sedge shoots and moss from the High T site were both significantly depleted in $^{13}C$ compared to Low T

site (both $p < 0.01$, Fig. 1b).

Peat C/N ratio and isotopic composition was significantly affected by depth, site and to a lesser extent by PFTs (Fig. 2a, 2b).

Carbon to N ratio increased with depth and was higher at the High T site compared to the Low T site (Fig. 2a, Table 2). In the

upper 0-2 cm peat layer, C/N ratios and stable isotopes corresponded with values observed for moss (Fig. 1, Fig. 2). Isotopic

composition of peat increments was significantly different for depths and PFTs (Table 2). $\delta^{13}C$ and $\delta^{15}N$ of peat increased with

depth (Fig. 2b, 2c) and the enrichment in $^{13}C$ with depth was stronger for sedge-cores than for shrub-cores (i.e. $2.7 \pm 0.4$ ‰

for sedge-cores and $1.6 \pm 0.4$ ‰ for shrub cores). The largest enrichment in $\delta^{13}C$ between the uppermost and lowest depth

increment appeared in sedge-cores at the High T site (Fig. 2b; i.e., $3.5 \pm 0.5$ ‰).

### 3.2 Chemical composition of vascular plants and moss-dominated peat analysed by py-GC/MS

Groups of pyrolysis products are given in Table 3. In the peat samples, polysaccharides (Ps) contributed 56 % - 86 %, and

phenols (Ph) 7 % - 30 % to all identified pyrolysis products, while the contribution from lignin-derived products (Lg) was <

15 %. The PCA (Fig. 3) clearly showed differences in the chemical composition of the PFT tissues and peat.

Principal components (PC) 1 to 3 explained 75.6 % and hence the major part of the variance in py-GC/MS data. Plotting the

scores of PC1 versus PC2 (Fig. 3a) revealed that plant materials of the PFTs as well as peat from different depths clustered

based on their pyrolysis products and thus on chemical composition. PC1 separated both shoot and root samples of shrubs and

sedges with exclusively positive scores from moss samples with exclusively negative scores. Thus, PC1 distinguished vascular plant samples from moss samples. Peat samples were arranged in between reflecting the contribution of both vascular plants

and mosses. The pyrolysis products responsible for the separation of PFTs are reflected in the factor loadings. Most phenols, including the parameter for sphagnum acid (4-isopropenylphenol, Ph6), and some polysaccharides (Ps1, 2, 4, 8, 9) had negative loadings on PC1, corresponding to moss samples (Fig. 3b). Lignin-derived products (sum of guaiacyl and syringyl lignin products, GS) had positive loadings and are indicative for shoot and root tissues of vascular plants (Fig. 3b).

Scores of PC2 separate aboveground shoot tissues of the three PFTs and the surface peat (negative scores) from root tissues

and the deeper peat increments (positive scores; Fig. 3a); within the peat samples, the deepest increments had the highest positive scores. Levoglucosan (Ps16) and the ratio of levoglucosan to the sum of polysaccharides (L/Ps) had the largest positive loadings, which indicates its relative enrichment with depth.

PC3 separates samples based on the two vascular plants, i.e. shrubs and sedges (Fig. 3c). Shrub shoot and root samples had exclusively negative scores. In contrast, all sedge shoot and root samples had positive scores. The peat samples clustered in-

between (i.e. low scores), but maintained the separation according to vascular plants. Likewise, the parameters selected to reflect the contribution from shrub, i.e. sum of $n$-alkenes and $n$-alkanes (Al, Table 1) and the ratio of $C_3$-guaiacol to the sum of guaiacyl products ($C_3$G/G, Table 1) had high negative loading on PC3. The parameter for sedge (the ratio of 4-vinylguaiacol to the sum of guaiacyl products (4VG/G), Table 1), showed positive loading on PC3 (Fig. 3b). The loadings furthermore suggest that both benzenes and levomannosan (Ps15) were associated to shrub as well, while 2,3-dihydro-5-methylfuran-2-

one (Ps3) and guaiacol (Lg2) showed high positive loading and are thus indicative for sedge in this context.

## 4 Discussion

### 4.1 Properties of a moss-dominated peat influenced by vascular plants

Testing our hypotheses requires the identification of properties being typical of the different vascular plants. In our study, we were able to apply a set of properties with a different degree of specificity to reveal how shrubs and sedges affect chemical

properties of moss-dominated peat.

#### 4.1.1 Different chemical properties of plant materials

Properties as C/N ratios, $\delta^{13}C$ and $\delta^{15}N$ of living plant parts showed significant differences between different PFTs (Fig. 1). C/N ratios of sedge shoots were significantly lower than those of shrubs or mosses which is in line with its higher decomposability (Kaštovská et al., 2018; Limpens and Berendse, 2003). $^{13}C$ signature of plant tissue tends to be depleted under warmer and drier conditions (Broder et al., 2012; Gavazov et al., 2016; Skrzypek et al., 2007) explaining the observed $\delta^{13}C$ depletion of sedge and moss tissues at the High T site compared to Low T site. Shrub shoots were significantly depleted in $\delta^{15}N$ compared to sedge and moss shoots (p < 0.001, Fig. 1c). The depletion of $\delta^{15}N$ in living *C. vulgaris* shoots compared to shoots of *E. vaginatum* and *Sphagnum spp.* is likely related to the symbiotic relationship between *C. vulgaris* and ericoid mycorrhizal fungi (Aerts et al., 2009; Bragazza et al., 2010; Emmerton et al., 2001). The transferred N from mycorrhiza to shrubs occurs to be depleted relative to soil N (Gebauer and Dietrich, 1993). Nevertheless, though differences in $\delta^{13}C$ and $\delta^{15}N$ between PFT existed, unknown variables influencing $\delta^{13}C$ and $\delta^{15}N$ in peat, such as various plant species growing in the peat, N deposition history etc. complicated the use of $\delta^{13}C$ and $\delta^{15}N$ as vegetation proxy.

By applying PCA to pyrolysates of plant and peat samples, we were able to clearly separate mosses from vascular plants and shrubs from sedges. The pyrolysis product specific for *Sphagnum* (4-isopropenylphenol) and the sum of lignin-derived products indicating vascular plants were particularly useful (Fig. 3a, 3b). Furthermore, the ratio of 4-vinylguaiacol to the summed guaiacyl products (4VG/G) indeed could be used to reflect sedges (Fig. 3c, 3d). Additionally, the ratio of C$_3$-guaiacol to the summed guaiacyl products (C$_3$G/G) and the sum of *n*-alkenes and *n*-alkanes (Al) could be confirmed as parameters indicative for shrubs (Fig. 3c, 3d).

#### 4.1.2 Effects of vascular plants on chemical properties of the moss-dominated peat

The high contribution of polysaccharides and phenols to peat pyrolysates (Table 3) and a strong positive correlation of the marker for sphagnum acid (4-isopropenylphenol) with the summed phenols (adj. $R^2$ = 0.98; Fig. 5a) indicates that the peat is dominated by *Sphagnum* tissue. The close positions of both peat samples and living *Sphagnum* shoots along PC1 visualise the dominance of *Sphagnum* in the peat. Furthermore, the similarity of C/N ratios, $\delta^{13}C$ and $\delta^{15}N$ of the uppermost peat increment to those of moss are indicative for moss-dominated peat (Schaub and Alewell, 2009) and has been measured likewise in *Sphagnum* peatlands by Kracht and Gleixner (2000). Thus, *Sphagnum* tissues dominate the peat of our two sites. Nevertheless,

py-GC/MS indicates that the moss-dominated peat has a contribution from vascular plants as lignin-derived products contributed up to 15 % to pyrolysates, particularly in the deeper increments under sedge coverage (Fig. 4a). Peat composition under shrubs and sedges is influenced by these species in the studied peat (0-20 cm) as indicated by the molecular parameters for sedge and shrub in the corresponding peat cores (Fig 4b, 4c, 4d). The sedge parameter (i.e. the ratio of 4-vinylguaiacol to the sum of guaiacols; Table 1) was higher in sedge-core samples than in shrub-core samples (Fig. 4b) illustrating a noticeable contribution from sedges to the peat. The correlation between the sedge parameter and the sum of lignin-derived products (adj. $R^2$ =0.52, Fig. 5c) indicates a strong contribution from sedges on lignin in sedge-cores. The contribution of shrubs to peat composition, as indicated by the suggested shrub parameter (i.e. the sum of $n$-alkenes and $n$-alkanes; Table 1), seems to be particularly high in the upper three peat increments as this parameter was mostly higher for shrub-core samples than for sedge-core samples in those increments (Fig. 4d). The increasing C/N ratios with depth at both sites could indicate a larger contribution of roots to the moss-dominated peat, since roots had much higher C/N ratios than shoots and moss (Fig. 1, 2a). By using py-GC/MS we could show that the peat was dominated by *Sphagnum*, and had a contribution from vascular plants; furthermore, we could validate the selected parameters described in the literature to reflect the contribution from sedge and shrub at the two sites.

**4.2 Decomposition of the moss-dominated peat**

Most of the studied parameters changed with depth in the peat cores and might be indicative for the assumed increase in peat decomposition with depth, i.e. with increasing time of exposure to oxygen after peat formation. In all peat cores, $\delta^{13}$C and $\delta^{15}$N increased with depth. Since $\delta^{13}$C and $\delta^{15}$N showed a positive correlation (adj. $R^2$ =0.48, Fig. 5b), their increases with depth may be caused by the same processes. The preferential uptake of lighter isotopes ($^{12}$C, $^{14}$N) for respiration by aerobic decomposers during decomposition causes a relative enrichment of heavier isotopes ($^{13}$C, $^{15}$N) in the remaining organic matter (Krüger et al., 2014, 2015; Nadelhoffer and Fry, 1988; Schaub and Alewell, 2009), suggesting that $^{13}$C and $^{15}$N depth trends are due to decomposition. Because $\delta^{15}$N trends may also be superimposed by N deposition (Novák et al., 2014) and fractionation processes during N fixation (Novák et al., 2016), $\delta^{13}$C seems to be a better indicator for peat decomposition in our study. Nevertheless, also $\delta^{13}$C peat records might be superimposed by differences in $\delta^{13}$C between shoots and roots, plant species, or site specific differences in $^{13}$C discrimination (Sect. 4.1.1). Differences in the C/N ratios did not provide a consistent

picture regarding changes in peat decomposition. We speculate that the observed increasing C/N ratio with depth might reflect an increasing N deposition in the past decades (Galloway et al., 2008) and an increased contribution of roots (high C/N ratios, Fig. 1a) to peat formation with increasing depth (Sect. 4.1.2). N deposition at the High T site were reported to be 8.2 kg ha$^{-1}$ y$^{-1}$ (Bragazza et al., 2005) and for peatlands in norther Italy between 4.2 and > 10 kg ha$^{-1}$ y$^{-1}$ (Bragazza et al., 2003, 2005).

In addition to changes in $\delta^{13}$C and $\delta^{15}$N reflecting the decomposition of the bulk peat (i.e. cumulative effects on all peat components), we examined changes in compounds being indicative for the decomposition of specific plant tissues, i.e., *Sphagnum*-derived peat (4-isopropenylphenol). Sphagnum acid and its pyrolysis product 4-isopropenylphenol have been found to very sensitively reflect aerobic decomposition of *Sphagnum* tissue (Abbott et al., 2013; Schellekens et al., 2015b). The observed decrease of 4-isopropenylphenol with depth occurred in all four peat cores (Fig. 4e) and its negative correlation with

$\delta^{13}$C (adj. $R^2$=0.27, Fig. 5d), confirms the increase in peat decomposition with depth as indicated by $^{13}$C.

The detected increase in polysaccharides with depth (Table 3) likely reflects the relative accumulation of rather resistant polysaccharides of *Sphagnum* cell walls during aerobic decomposition (Hájek et al., 2011) and the preferential decomposition of *Sphagnum* phenols (Schellekens et al., 2015b). However, the polysaccharide products from the peat samples can have multiple sources, too (e.g. *Sphagnum* cell walls, or ligno-cellulose from vascular plants; Sarkar et al., 2009). Similarly, the

increase with depth of the ratio of levoglucosan to the sum of polysaccharide-derived products (L/Ps) may reflect the relative preservation of *Sphagnum* polysaccharides during aerobic decomposition (Table 1, Fig. 4f). This is confirmed by a strong negative correlation between this ratio and 4-isopropenylphenol (adj. $R^2$=0.79, Fig, 5e), and further by the positive loading of L/Ps on PC2.

### 4.2.1 Effects of vascular plants on decomposition of moss-dominated peat

We found three indicators that decomposition of moss-dominated peat with depth is boosted by sedge coverage compared to shrub coverage. Increases in $\delta^{13}$C with depth were higher for sedge-cores than for shrub-cores (Fig. 2b) and the decrease with depth of 4-isopropenylphenol was mostly stronger in sedge-cores than shrub-cores (Fig. 4e). In general, the L/Ps ratio increased with depth (Fig. 4f). Only in the deepest increment of sedge-cores this ratio decreased. This decrease could indicate that decomposition under sedge coverage in the deepest peat increment is so high that even less decomposable polysaccharides

have been decomposed. Because this is not evident from the 4-isopropenylphenol record, it probably reflects a higher contribution of sedge-derived polysaccharides at these depths.

The observed decomposition patterns were detected by a parameter describing the whole peat ($\delta^{13}$C), but were also reflected by compounds indicative for *Sphagnum* material (4-isopropenylphenol). The latter suggests that (changes in) vascular plant composition may affect the decomposition of the existing *Sphagnum* peat by changing plant-soil feedbacks (Bragazza et al., 2013; Gavazov et al., 2018; Robroek et al., 2015). The observed higher degree of degradation of peat under sedges than under shrubs may be explained by differences in litter quality or root traits. Sedge litter is likely to be more readily decomposable compared to shrub litter, caused by its lower C/N ratios (Fig. 1a; Huang et al., 1998; Kaštovská et al., 2018; Laiho et al., 2003; Limpens and Berendse, 2003). Furthermore, additional oxygenation by the aerenchym of *E. vaginatum* might trigger further decomposition of moss-dominated peat (Armstrong, 1964; Holzapfel-Pschorn et al., 1986; Roura-Carol and Freeman, 1999). This process is particularly relevant at the Low T site, where the uppermost 20 cm of the peat remained water saturated much longer than at the High T site.

**4.2.2 Temperature effects on decomposition of the moss-dominated peat – interactions with vascular plant effects**

The altitudinal gradient has been used to reveal potential effects of increasing temperature and associated lower water table on peat decomposition by comparing the suggested decomposition parameters (Table 1) between the High T and Low T site. Increases in $\delta^{13}$C with depth were higher at the High T site than at the Low T site (Fig. 2). Therefore, decomposition of the moss-dominated peat is likely to be increased at the High T site compared to the Low T site independent of the vascular plant species.

However, depth trends of the *Sphagnum*-specific decomposition parameter (4-isopropenylphenol) do not reflectthis increased peat decomposition at higher temperatures. 4-Isopropenylphenol decreased less at the High T site compared to the Low T site (Fig. 4e). We might speculate a difference in temperature sensitivity between decomposition of *Sphagnum* and the whole peat, but such a hypothesis needs to be tested by e.g. temperature controlled incubation experiments.

The ratio of levoglucosan to the sum of polysaccharides (Fig. 4f) did not show consistent trends related to temperature again indicating no temperature enhanced degradation of the moss-dominated peat. Vascular plants and particularly sedges might increasingly contribute to polysaccharides with higher temperatures (see section 4.2.1). These changes from more *Sphagnum-*

derived polysaccharides to more sedge-derived polysaccharides could change decomposition dynamics of polysaccharides, since cell walls constituents from *Sphagnum* were found to be less easily decomposable (Hájek et al., 2011). A higher contribution from sedges could therefore superimpose a potential enrichment of *Sphagnum*-derived polysaccharides with depth. On the other hand, it seems unlikely that the observed depth trends for the *Sphagnum* specific 4-isopropenylphenol has been affected in a similar way as the less specific ratio of levoglucosan to the sum of polysaccharides. Given the above, the

higher degree of peat decomposition at the High T site picked up by $\delta^{13}C$ is probably mediated by higher input rates of easily decomposable vascular plant litter, notably sedges, combined with the warmer and drier conditions favouring aerobic decomposition processes (Biester et al., 2014).

This combined effect of sedges and temperature on peat decomposition has implications for the long-term C storage in *Sphagnum*-dominated peatlands because of projected shifts from sedges to shrubs with climate change (Breeuwer et al., 2009).

That change towards less sedges may partly offset temperature driven decomposition processes because of the observed enhancing effect of sedges on peat decomposition. Less sedges (i.e. more shrubs) should result in less peat degradation because (i) the decomposability of available litter (higher contribution of shrub litter) is reduced as its chemical composition indicates less decomposability (Kristensen and McCarty, 1999; Ward et al., 2015), (ii) an increasing presence of shrubs (*C. vulgaris*) supresses belowground biota activity and nutrient cycling (Fenner and Freeman, 2011), (iii) *C. vulgaris* associates with

mycorrhizal fungi which increase the uptake of organic nutrients leading to an increase of C/N ratio (Read et al., 2004) and thus a decrease in peat decomposition (Ward et al., 2015), (iv) the input of labile C into peat via sedge roots is lower (Crow and Wieder, 2005; Robroek et al., 2015), (v) the transport of oxygen into peat via aerenchym of *E. vaginatum* is lower. On the other hand, Zeh et al. (2019) could show that shrubs translocated more C into the peat at higher temperatures than sedges, which could result in reinforcing effect on peat decomposition with increasing temperature. Obviously, the enhanced C input

by shrubs into peat did not coincide with enhanced decomposition of the moss-dominated peat on these sites.

**5 Conclusions**

The studied plant functional types differed in the chemical composition of their biomass, and therefore in litter quality. Although both peatlands were moss-dominated, the application of several complementary parameters revealed clear influences from sedge and shrub litter. Combining data obtained by py-GC/MS and isotopic analysis enabled separating the effects of

PFT and temperature on peat decomposition under field conditions. Whereas changes in $\delta^{13}C$ depth records reflected the state of peat degradation, its application to disentangle the effects of source material, decomposition processes, and environmental factors is incomplete. The C/N ratio and $\delta^{15}N$ were not specific enough to represent vascular plant effects on decomposition of the moss-dominated peat. Combining records of molecular parameters and $\delta^{13}C$ indicated that moss-dominated peat was more decomposed under sedge than under shrub coverage, particularly under high temperatures. The most important and also surprising result of our study was that vascular plants had a more pronounced impact on peat decomposition than temperature and associated lower water tables together. Potential $O_2$ transport by the aerenchym of sedges did probably not contribute to enhanced peat decomposition at the High T site, as the top 20 cm peat layer sampled remained above the water table, and thus aerated, for most of the year. Considering that climate change can lead to a shift from *Sphagnum* mosses to vascular plants and from sedges to shrubs (Breeuwer et al., 2009), an increase of sedge coverage may enhance the decomposition of *Sphagnum* peat in surface layers at elevated temperature. To what extent this increased decomposition may be partly compensated by growing dominance of shrubs over sedges with climate change deserves further studies in order to link belowground decomposition to aboveground production.

**Data availability**

The underlying py-GCMS data can be accessed via http://dx.doi.org/10.25532/OPARA-77. Carbon to nitrogen (C/N) ratios, $\delta^{15}N$ [‰] and $\delta^{13}C$ [‰] of shoot and root tissues from the three peat forming plants and of peat increments are available via http://dx.doi.org/10.25532/OPARA-78 and http://dx.doi.org/10.25532/OPARA-79, respectively.

**Author contributions**

JL, LB and KK designed the study and MTI, LZ, JL, LB and KK collected the samples and data in the field. MTI and LZ processed the samples and did the analyses. JS contributed to the py-GCMS part of the article, including analysis of the data and editing on the paper. LZ took the lead in preparing the manuscript, with contributions from all co-authors.

## Competing interests

The authors declare that they have no conflicts of interest.

## Acknowledgement

We thank the Ufficio Ecologia del Paesaggio - Provincia Autonoma di Bolzano and the Servizio Sviluppo Sostenibile e Aree

Protette - Provincia Autonoma di Trento for access to the Italian sites; Manuela Unger and Gisela Ciesielski for sample

preparation and analyses of C, N and stable isotopes in the laboratory; Josephine Hillig for carrying out the py-GC/MS

measurements. For financial support we thank TU Dresden, Wageningen University and University of Ferrara (FAR 2014),

FAPESP projects 2013/03953-9 and 2016/03337-4. The py-GC/MS system was funded by DFG project INST 269/575-1.

## Review statement

This paper was edited by Sébastien Fontaine and reviewed by Tim Moore and an anonymous referee.

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

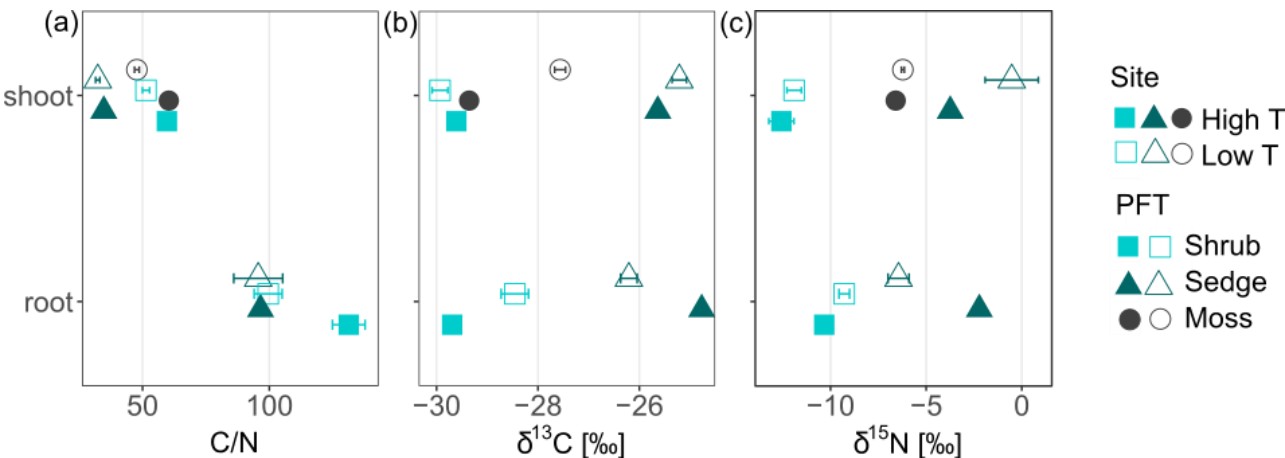

**Figure 1: Carbon to nitrogen (C/N) ratios, δ¹⁵N [‰] and δ¹³C [‰] of shoot and root tissues from three plant functional types (PFTs; i.e. mosses, shrubs and sedges) and from two peatlands differing in temperature (high and low temperature indicated by High T and Low T, respectively). Symbols for shoots represent mean values with standard error (*n* = 5), whereas symbols for roots give means of two or a single value (High T sedge root).**


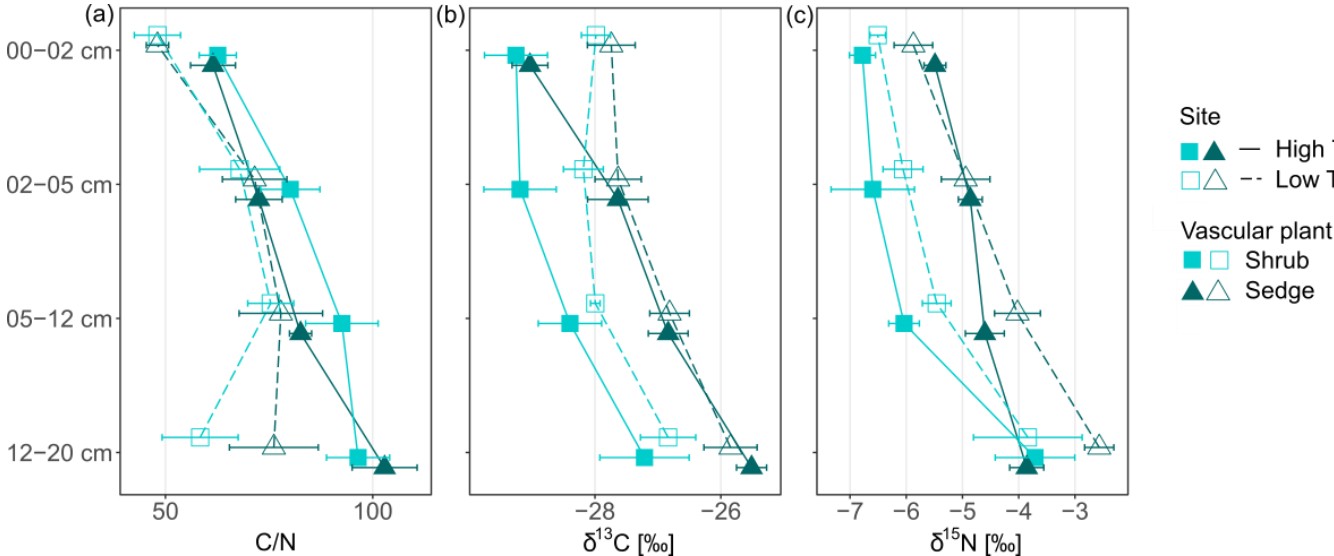

**Figure 2: Carbon to nitrogen (C/N) ratios, δ¹⁵N [‰] and δ¹³C [‰] of peat core increments from two peatlands differing in temperature (high and low temperature indicated by High T and Low T, respectively) and covered by two different vascular plants (shrub and sedge). Symbols represent mean values with standard error (*n* = 5).**


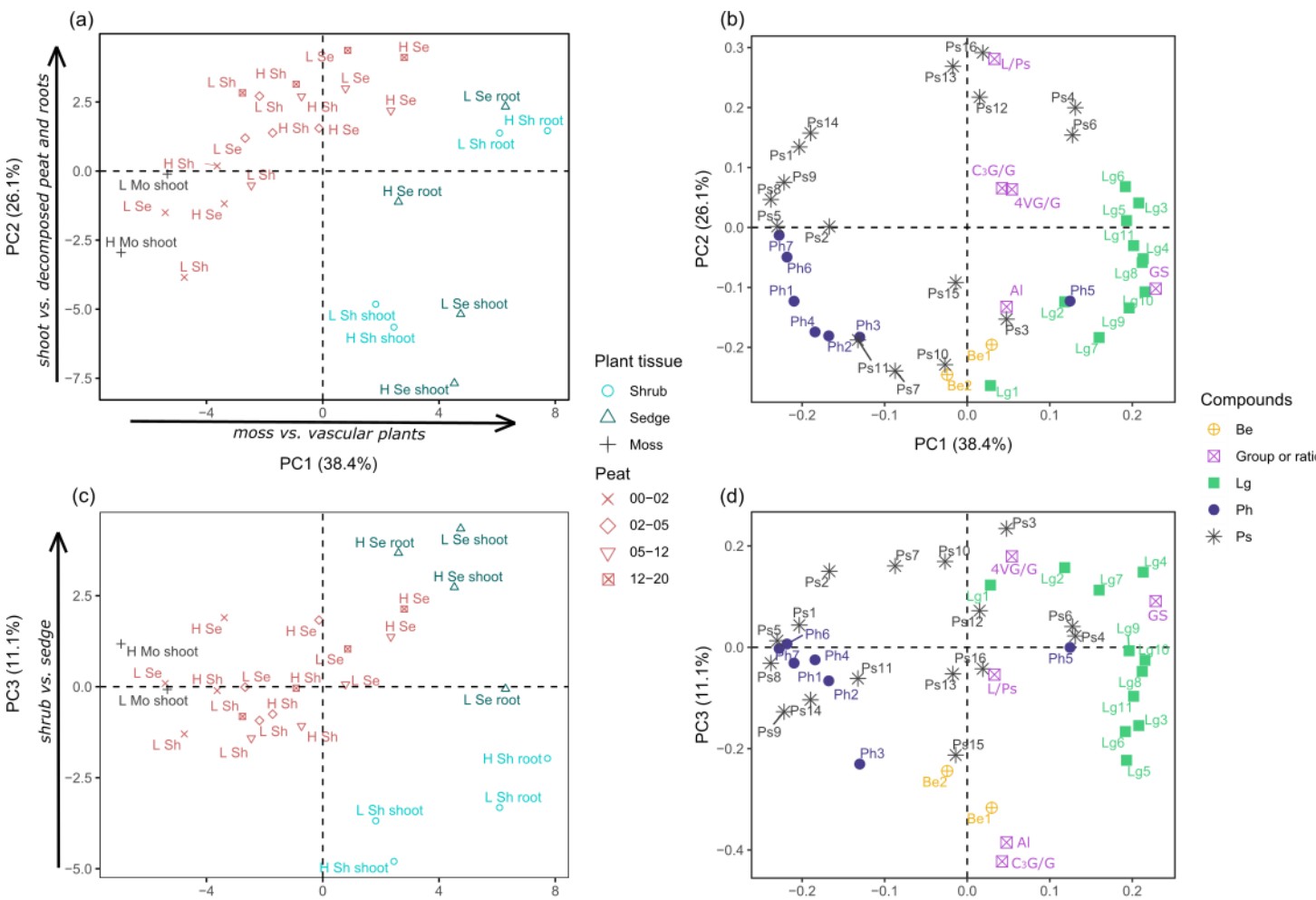

**Figure 3: Results of the principal component analysis (PCA) of the pyrolysis data to reveal molecular composition of peat and peat forming plants. PC1 to PC2 projections of scores (a) and loadings (b), and PC1 to PC3 projections of scores (c) and loadings (d). Plant samples are labelled by site (H = high temperature; L = low temperature) and plant tissue (shoot, root). Peat samples are named by site and vascular plants (Se = Sedge-peat; Sh = Shrub-peat). The four depth increments are distinguished by symbols. Pyrolysis products correspond to codes that are given in (A1). Abbreviations are according to chemical group: benzenes (Be), lignin products (Lg), phenols (Ph) and carbohydrates (Ps). Additional variables are included into PCA, namely summed $n$-alkenes and $n$-alkanes (Al), 4-vinylguaiacol to the summed guaiacyl products (4VG/G), $C_3$-guaiacol to the summed guaiacyl products ($C_3$G/G), sum of guaiacyl and syringyl lignin products (GS), levoglucosan to summed polysaccharides (L/Ps).**


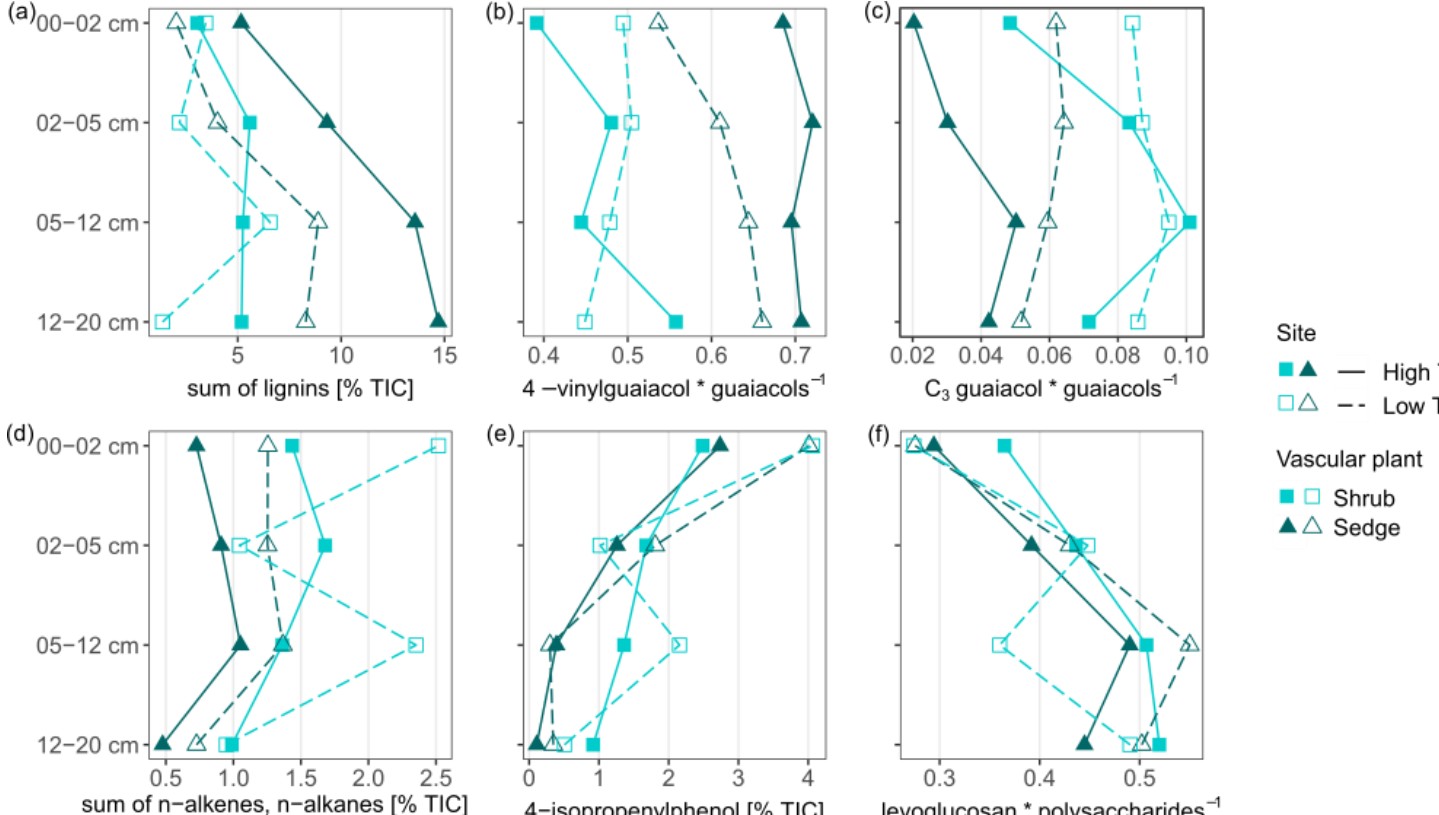


**Figure 4: Parameters that reflect vascular plant input in moss-dominated peat (a-d) and decomposition of the moss-dominated peat (e, f) in peat core increments from both sites and given peat cores. Data derived from pyrolysis gas chromatography/mass-spectrometry**

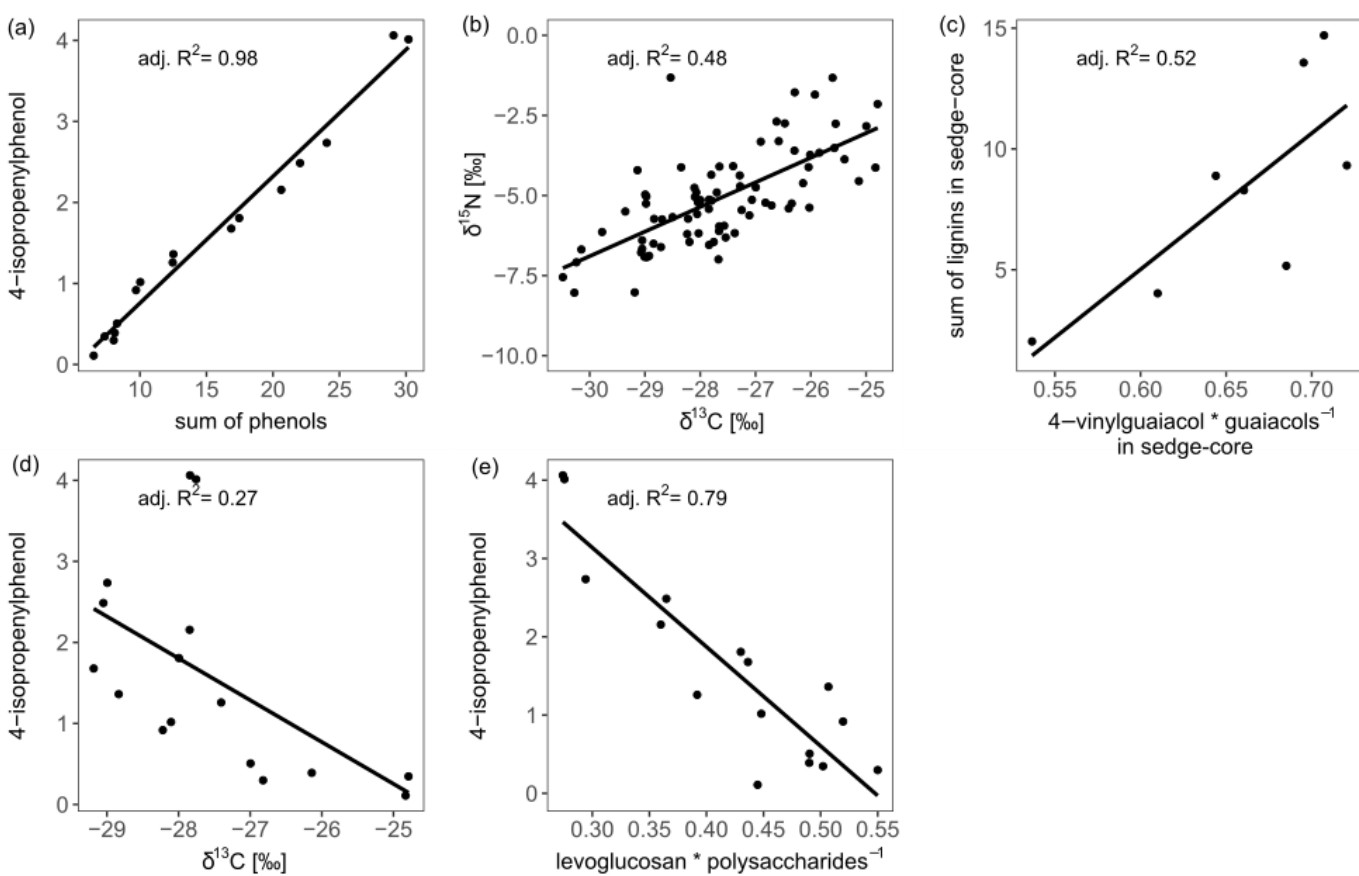

**Figure 5: Regression plots between parameters describing decomposition of the moss-dominated peat and the impact of vascular plants on peat properties.**



**Table 1: Overview of all parameters used in this study to determine impacts of plant functional types (PFT) on peat properties and decomposition of the moss-dominated peat (for abbreviations of the parameter see legend of Fig. 3).**

| Parameter | Unit | Indication | Interpretation in *Sphagnum*-dominated peat |
|---|---|---|---|
| C/N | - | preferential decomposition of C over N | aerobic decomposition |
| $\delta^{13}$C | [‰] | preferential decomposition of $^{12}$C over $^{13}$C isotope | aerobic decomposition |
| $\delta^{15}$N | [‰] | preferential decomposition of $^{14}$N over $^{15}$N isotope | aerobic decomposition |
| sum of G and S lignin products | [% TIC[a]] | lignin | vascular plants (van Smeerdijk and Boon, 1987) |
| sum of *n*-alkenes and *n*-alkanes | [% TIC[a]] | cutan, suberan, leaf waxes | ericoid shrubs (Schellekens and Buurman, 2011; van Smeerdijk and Boon, 1987) |
| C$_3$G/G | - | intact lignin | ericoid shrubs (Schellekens et al., 2012) |
| 4-vG/G | - | ferulic acid | sedges (van Smeerdijk and Boon, 1987; Schellekens et al., 2012) |
| 4-isopropenylphenol (Ph6) | [% TIC[a]] | sphagnum acid | aerobic decomposition of *Sphagnum* tissues (preferential loss of polyphenols over polysaccharides; Schellekens et al., 2015b) |
| levoglucosan/sum of polysaccharides | - | cellulose | aerobic decomposition of *Sphagnum* tissues (preservation of *Sphagnum* polysaccharides; Schellekens et al., 2015b)[b] |

[a] total ion current
[b] interpretation refers to relatively low values

**Table 2: Type II analysis of variance with Kenward-Roger's approximation of degree of freedoms applied on carbon to nitrogen (C/N) ratios, $\delta^{13}C$ and $\delta^{15}N$ of peat cores ($n = 5$).**

| | C/N | | | $\delta^{13}C$ | | | $\delta^{15}N$ | | |
|---|---|---|---|---|---|---|---|---|---|
| | DF | F-test | p | DF | F-test | p | DF | F-test | p |
| Vascular plants (VP) | 1 | 0.15 | 0.69 | 1 | 27.05 | **< 0.001** | 1 | 25.79 | **< 0.001** |
| Site | 1 | 19.04 | **0.002** | 1 | 3.17 | 0.11 | 1 | 1.62 | 0.24 |
| Depth | 3 | 12.75 | **< 0.001** | 3 | 24.21 | **< 0.001** | 3 | 28.46 | **< 0.001** |
| VP:Site | 1 | 1.51 | 0.22 | 1 | 1.80 | 0.18 | 1 | 0.01 | 0.94 |
| VP:Depth | 3 | 0.97 | 0.41 | 3 | 1.90 | 0.13 | 3 | 0.94 | 0.43 |
| Site:Depth | 3 | 2.40 | *0.07* | 3 | 2.07 | 0.11 | 3 | 0.51 | 0.68 |
| Site:Depth:VP | 3 | 0.12 | 0.94 | 3 | 0.32 | 0.81 | 3 | 1.26 | 0.29 |

**Table 3: Groups of all identified pyrolysis products for all samples: *n*-alkenes and *n*-alkanes (Al), benzenes (Be), lignin products (Lg), phenols (Ph) and polysaccharides (Ps). The contribution of the single products in groups to total ion current were cumulated for each of the respective group.**

| Sample | PFT | Site | Al | Be | Lg | Ph | Ps |
|--------|-----|------|-----|-----|-----|-----|-----|
| **Plant tissue** | | | | | | | |
| Shoot | Sedge | Low T | 0.78 | 2.31 | 46.57 | 7.63 | 42.70 |
| Shoot | Sedge | High T | 1.11 | 4.87 | 48.89 | 15.42 | 29.71 |
| Shoot | Shrub | Low T | 3.98 | 6.63 | 26.40 | 21.87 | 41.12 |
| Shoot | Shrub | High T | 5.41 | 8.72 | 30.09 | 19.15 | 36.63 |
| Shoot | Moss | Low T | 1.13 | 2.83 | 6.25 | 25.37 | 64.42 |
| Shoot | Moss | High T | 1.02 | 3.37 | 8.76 | 33.39 | 53.46 |
| Root | Sedge | Low T | 0.77 | 1.85 | 32.23 | 5.50 | 59.65 |
| Root | Sedge | High T | 0.85 | 1.28 | 26.57 | 14.09 | 57.21 |
| Root | Shrub | Low T | 3.53 | 3.26 | 23.42 | 8.90 | 60.89 |
| Root | Shrub | High T | 2.19 | 1.97 | 24.71 | 7.35 | 63.78 |
| **Peat** | | | | | | | |
| 00-02 cm | Sedge | Low T | 1.25 | 3.49 | 9.18 | 30.18 | 55.89 |
| 00-02 cm | Sedge | High T | 0.73 | 2.63 | 14.98 | 24.04 | 57.62 |
| 00-02 cm | Shrub | Low T | 2.52 | 5.64 | 13.65 | 29.06 | 49.13 |
| 00-02 cm | Shrub | High T | 1.44 | 2.86 | 8.77 | 22.03 | 64.90 |
| 02-05 cm | Sedge | Low T | 1.25 | 2.28 | 8.99 | 17.46 | 70.02 |
| 02-05 cm | Sedge | High T | 0.91 | 1.89 | 18.10 | 12.46 | 66.65 |
| 02-05 cm | Shrub | Low T | 1.04 | 1.94 | 4.60 | 10.03 | 82.39 |
| 02-05 cm | Shrub | High T | 1.68 | 2.31 | 9.45 | 16.86 | 69.70 |
| 05-12 cm | Sedge | High T | 1.05 | 1.63 | 21.69 | 8.11 | 67.52 |
| 05-12 cm | Sedge | Low T | 1.37 | 2.00 | 12.27 | 8.03 | 76.33 |
| 05-12 cm | Shrub | Low T | 2.35 | 3.07 | 12.72 | 20.62 | 61.24 |
| 05-12 cm | Shrub | High T | 1.36 | 2.34 | 7.87 | 12.51 | 75.91 |
| 12-20 cm | Sedge | Low T | 0.73 | 1.15 | 11.31 | 7.35 | 79.46 |
| 12-20 cm | Sedge | High T | 0.48 | 0.85 | 20.23 | 6.52 | 71.92 |
| 12-20 cm | Shrub | Low T | 0.95 | 1.59 | 2.65 | 8.28 | 86.53 |
| 12-20 cm | Shrub | High T | 0.99 | 1.54 | 8.19 | 9.70 | 79.58 |

685

690

**(A1) Quantified pyrolysis products, their codes, retention time (RT), and specific ion fragments (*m/z*) used for quantification.**

| Code | Pyrolysis product | m/z | RT[c] |
|------|-------------------|-----|-----|
| - | $C_{19}$-$C_{28}$ *n*-alkenes | 55, 69 | - |
| - | $C_{19}$-$C_{29}$ *n*-alkanes | 57, 71 | - |
| Be1 | benzene | 78 | 2.365 |
| Be2 | toluene | 91, 92 | 3.86 |
| Lg1 | 4-vinylphenol | 91, 120 | 20.117 |
| Lg2 | guaiacol | 109, 124 | 15.232 |
| Lg3 | 4-methylguaiacol | 123, 138 | 19.092 |
| Lg4 | 4-vinylguaiacol | 135, 150 | 23.314 |
| Lg5 | $C_3$ guaiacol, *trans* | 164 | 27.73 |
| Lg6 | 4-acetylguaiacol | 151, 166 | 28.9 |
| Lg7 | syringol | 139, 154 | 24.621 |
| Lg8 | 4-methylsyringol | 153, 168 | 27.687 |
| Lg9 | 4-vinylsyringol | 165, 180 | 31.239 |
| Lg10 | $C_3$ syringol, *trans* | 194 | 35.049 |
| Lg11 | 4-acetylsyringol | 181, 196 | 35.987 |
| Ph1 | phenol | 66, 94 | 11.216 |
| Ph2 | $C_1$ phenol | 107, 108 | 14.031 |
| Ph3 | $C_1$ phenol | 107, 108 | 14.819 |
| Ph4 | $C_2$ phenol | 107, 122 | 18.241 |
| Ph5 | catechol | 110 | 19.536 |
| Ph6 | 4-isopropenylphenol | 119, 134 | 23.039 |
| Ph7 | *p*-hydroxybiphenyl | 170 | 35.33 |
| Ps1 | (2*H*)furan-3-one | 54, 84 | 4.566 |
| Ps2 | 2-furaldehyde | 95, 96 | 5.605 |
| Ps3 | 2,3-dihydro-5-methylfuran-2-one | 98 | 8.826 |
| Ps4 | unidentified carbohydrate | 55, 86 | 10.29 |
| Ps5 | 5-methyl-2-furaldehyde | 109, 110 | 10.328 |
| Ps6 | 4-hydroxy-5,6-dihydro-(2*H*)-pyran-2-one | 114 | 11.648 |
| Ps7 | 2-hydroxy-3-methyl-2-cyclopenten-1-one | 55, 112 | 12.861 |
| Ps8 | dianhydrorhamnose | 113, 128 | 13.324 |
| Ps9 | unidentified carbohydrate | 128, 72 | 15.057 |
| Ps10 | unidentified carbohydrate | 56, 114 | 16.52 |
| Ps11 | 1,4:3,6-dianhydro-α-D-glucose | 69, 57 | 19.617 |
| Ps12 | 1,4-Anhydroxylofuranose | 57 | 21.644 |
| Ps13 | 1,4-Dideoxy-D-glycero-hex-1-enopyranose-3-ulose | 87, 144 | 22.882 |
| Ps14 | levogalactosan | 60 | 25.359 |
| Ps15 | levomannosan | 60 | 27.862 |
| Ps16 | levoglucosan | 60 | 30.045 |