# Peer review of "Vascular plants affect properties and decomposition of mossdominated peat, particularly at elevated temperatures"

_Biogeosciences, 2019_

## Referee Comment (RC1) · Anonymous Referee #1 · 7 Feb 2020

The ms describes a study on the influence of changes in plant cover on the decomposition of moss (sphagnum) dominated peat at two peatlands differring in altitude and temperature. The authors investigated a large number of short cores ($\sim$ 20 cm) for changes in peat decomposition based on C/N ratios, $\delta$13C and changes in organic components derived from pyrolysis GC-MS analyses. The authors found out, that sedges and shrubs litter increases decomposition of moss dominated peat, especially at higher temperature. The effect of vascular plants was more pronounced that that of temperature. Peat decomposition appears to be best reflected by $\delta$13C, although its application to distinguish source material from decomposition is seen incomplete. C/N and $\delta$15N appear to be not specific enough to indicate the effects of vascular plants on

moss peat decomposition. The topic and the conclusions are not entirely new, but this is to my knowledge the most comprehensive study on this topic. Especially the combination of stable isotopes, C/N ratios and pyrolysis GC-MS gives new insights into the role of plant cover for initial organic matter degradation in peatlands on a molecular level. The ms is suitable for Biogeosciences and well written and I suggest publication after addressing some issues.

Specific comments: The use of the term decomposition in peatlands is often a bit unspecific and many different methods (compound specific or only operational defined) are used to determine the degree of peat decomposition. Although the authors describe in the ms what C/N, $\delta$13C and py-GC-MS can show, but it remains unclear when I see mass loss (C/N and $\delta$13C) or changes in the molecular composition. The problem here, I think is that e.g. C/N and $\delta$13C were used in previous studies do distinguish changes in peat decomposition (here it describes mainly mass loss (polysacarides), but what the authors investigated in their study is the initial phase of plant material decay (a lot of qualitative changes/ molecular composition) I suggest that the authors make clear what they mean by "peat" and give a clear statement e.g in their hypotheses what they mean by "decomposition". The authors address the importance of oxygen availability, redox conditions and water levels at the time of sampling. They also mention that oxygen transport into the peat via aerenchym of E. vaginatium takes place. However, redox-conditions (here the availability of oxygen for OM mineralisation) are crucial for peat decomposition. For me it remains unclear how much of the observed changes in decomposition/OM quality are related to redox-conditions/water table depth or oxygen transport via the roots of vascular plants and how much to the presence of pant litter from sedges or shrubs. More shrubs and sedges in peatlands are usually a result of drier conditions. Drier conditions mean lower mean water table and aeration/increased decomposition of peat, a bit a hen and egg problem. The authors remain unclear about this in their conclusions.

- The authors tend to use general terms such as L 35 alterations in the environment,...

L39 plant-soil feedbacks, .L66..hydrological conditions. Please be more specific. - Can the authors give an estimate about the ages of their plant/peat samples. Is time an issue here? - L185-195 The description of Fig. 1 2 and 4 are a bit difficult to decipher. May be disniguish by site. - Table 1: all reference are from one of the authors (Schellekens). Any independent references available here?

---

## Referee Comment (RC2) · Tim Moore (Referee) · 20 Feb 2020

This manuscript examines the influence of variations in plant cover on the rate of decomposition in the upper layers of the peat profile, at two sites in the Italian Alps, which vary in their mean annual temperature. The aim is to provide some indication of what may happen if climate change warms peatlands and the vegetation cover of sedges (here Eriophorum) and shrubs (here Calluna) increases at the expense of Sphagnum moss. Peat cores were analyzed for a wide range of properties, related to degree of decomposition, including pyrolysis, which is unusual in peat studies. The results suggested that both temperature (over a 1.4oC range) and vegetation cover influenced

decomposition rate of the peat, dominated by residual moss, but that changes in vegetation to sedges and to a lesser shrubs, were more important than the temperature rise, using the two sites as proxies for change.

This contribution is one of several suggesting that changes in vegetation from global change are likely to be more important than simple rises in temperature in affecting the C budget of ecosystems, for example the 'shrubification' of the Arctic. Here, detailed and careful sampling of two sites, with modest differences in air temperature but varying in vascular plant coverage 47 and 77%), provide a suitable analogue to address this issue. The peat samples, and vegetation, have been analyzed by a variety of techniques, some of them common, such as elemental and stable isotopes, and some less common, such as gc/ms pyrolysis. The content of the manuscript is suitable for Biogeosciences and it is generally well written, though I have noted a few errors of the pdf, along with some specific comments.

Some comments for 'discussion':

The sites vary in terms of their mean annual temperature, but is this translated to similar differences in the peat layer undergoing decomposition? Are soil temperature data available to be more precise on the thermal differences in the peat at the two sites? It might be smaller or larger than the 1.4oC.

Is there an estimate at the rate of peat accretion at the sites? In other words can you estimate over what period the 20 cm of peat have accumulated (e.g. by 210Pb dating, perhaps a century?) and what are the changes in the environment over that period? Is what we see now, the same as what it was a century ago, when the current 20 cm peat began to form? For line 263, can you provide an estimate of 'increasing time of exposure'?

Do you have any estimates of the input of litter into the sites, based on the vegetation composition, to provide a quantitative context of 'how much' is being added? The references cited (lines 41-42) tended to be for Arctic tundra, which is presumably inapplicable to alpine conditions.

I think that careful attention should be given to the water table at the two sites which are reported on line 89. Perhaps the Zeh et al. (2019) ms contains more specific information, but a difference in water table of 30 cm (the minimums reported) would have a profound effect on decomposition rates in the peat cores, the High T site being both warmer and drier . . . . . . . Was August 2015 to July 2016 'typical' in terms of hydrology (i.e. precipitation etc.)? On the other hand, if the highest water table measured was 17 and 15 cm, it means that only the bottom 3 to 5 cm of the 20 cm core were at and under the water table, so we are dealing with decomposition under aerobic conditions, effectively the acrotelm. Perhaps a useful metric would be the proportion of the year in which the water table was within the 20 cm core, especially the 15-20 cm section, to see whether hydrology was significantly different at the two sites. An increased vascular cover, associated with a warming, will likely increase evapotranspiration rates, which in turn will produce a lowered water table, accelerating the vascular 'invasion'.

Eriophorum is arenchymous, with the capacity to oxygenate the peat: would that influence the peat environment in terms of decomposition rate, given that the top 20 cm is above the water table for most of the year?

Line 190: I was surprised to see the large increase in C:N ratio with depth in nearly all the cores, whereas with decomposition one might have expected a decline in the ratio. Is there an explanation for this pattern – I could not see one in the Discussion (cf Table 1). Does atmospheric N deposition play a role here (larger N concentrations in the past few decades)?

Line 230 I would think that there are major differences in 15N among the three plant types from zero to -10, which relate back to, I assume, the mycorrhizal dependance of Calluna, the non-mychorrizal Eriophorum and moss in between

I got goggle-eyed looking at the symbols in Figures 1, 2 and 4 and would appreciate some differentiation stronger than washed-out blue and a yukky looking green. Be

'artistic'! Simple black and red would be nice. . . . . .

4.2.2 is a 45 line 'paragraph' and it might be easier to digest if it was broken down into three paragraphs, each dealing with a specific theme. It is a 'confounding' system with multiple interpretations of results and the strength of the ms is the range of analyses conducted.

In the Conclusion, or somewhere in the Discussion, it would be useful to identify the 'bang for the buck' in these analyses: some are simple and routine and some, especially the gc/ms pyrolysis, is 'labour intensive'. Do you have anything to add to the Biester et al. 2014 paper, based on this specific application?

I provide a copy of the pdf which I have annotated with comments and suggested typographical and other correction.

Tim Moore

Please also note the supplement to this comment:
https://www.biogeosciences-discuss.net/bg-2019-503/bg-2019-503-RC2-supplement.pdf

**Supplement:**

[revised manuscript text omitted]

---

## Author Comment (AC1) · 7 Apr 2020

We highly appreciate the very helpful and constructive comments of the anonymous referee, which helped us to further improve the manuscript. We tried to consider all of them.

The referee's comments will we shown black. Our response is shown blue, *italic* and tab-indented while changes in the manuscript are in quotation marks and **bold**.

**Anonymous Referee #1**

The ms describes a study on the influence of changes in plant cover on the decomposition of moss (sphagnum) dominated peat at two peatlands differring in altitude and temperature. The authors investigated a large number of short cores (20 cm) for changes in peat decomposition based on C/N ratios, 13C and changes in organic components derived from pyrolysis GC-MS analyses. The authors found out, that sedges and shrubs litter increases decomposition of moss dominated peat, especially at higher temperature. The effect of vascular plants was more pronounced that that of temperature. Peat decomposition appears to be best reflected by 13C, although its application to distinguish source material from decomposition is seen incomplete. C/N and 15N appear to be not specific enough to indicate the effects of vascular plants on moss peat decomposition. The topic and the conclusions are not entirely new, but this is to my knowledge the most comprehensive study on this topic. Especially the combination of stable isotopes, C/N ratios and pyrolysis GC-MS gives new insights into the role of plant cover for initial organic matter degradation in peatlands on a molecular level. The ms is suitable for Biogeosciences and well written and I suggest publication after addressing some issues.

Specific comments: The use of the term decomposition in peatlands is often a bit unspecific and many different methods (compound specific or only operational defined) are used to determine the degree of peat decomposition. Although the authors describe in the ms what C/N, 13C and py-GC-MS can show, but it remains unclear when I see mass loss (C/N and 13C) or changes in the molecular composition. The problem here, I think is that e.g. C/N and 13C were used in previous studies do distinguish changes in peat decomposition (here it describes mainly mass loss (polysacarides), but what the authors investigated in their study is the initial phase of plant material decay (a lot of qualitative changes/ molecular composition) I suggest that the authors make clear what they mean by "peat" and give a clear statement e.g in their hypotheses what they mean by "decomposition".

> *Thank you for this suggestion. We will include a sentence what we understand as "decomposition". We will integrate this at the end of the sentence in L.55 as follows:*

"**We defined decomposition as any changes in properties of the peat relative to its source material, e.g. plant material from *Calluna vulgaris Eriophorum vaginatum*, *Sphagnum spp.* directly after deposition.**"

> *We will also include a definition of "peat" at the end of the sentence in L.54:*

"**Peat was defined as any organic material that accumulates underneath the peatlands surface, including living stems of *Calluna vulgaris*, *Eriophorum vaginatum* and *Sphagnum spp.***"

The authors address the importance of oxygen availability, redox conditions and water levels at the time of sampling. They also mention that oxygen transport into the peat via aerenchym of E. vaginatium takes place.
However, redox-conditions (here the availability of oxygen for OM mineralisation) are crucial for peat decomposition. For me it remains unclear how much of the observed changes in decomposition/OM quality are related to redox-conditions/water table depth or oxygen transport via the roots of vascular plants and how much to the presence of pant litter from sedges or shrubs. More shrubs and sedges in peatlands are usually a result of drier conditions. Drier conditions mean lower mean water table and aeration/ increased decomposition of peat, a bit a hen and egg problem. The authors remain unclear about this in their conclusions.

*We want to thank the referee for this valuable comment. We agree on the referee's comment that redox conditions are a major control on peat decomposition. This was considered in the sampling design, where we restricted the plot installation to hummocks and peat core sampling to the aerobic zone, the acrotelm. Also we specifically chose locations where sedges and shrubs grew in mixed stands, experiencing similar redox conditions. With this strategy we avoided the "hen and egg problem" as much as possible. However, as addressed to the review provided by Tim Moore, sampling depths in the peat cores are periodically water saturated. Water table measurements 2015/08–2016/07 (three gauges at each site) show that the peat within the sampled 20 cm on the High T site stayed aerated for 117, 360 or 366 days of the year as recorded in the three gauges. On the Low T site, the peat of the top 20 cm was aerated for 137, 138 or 284 days of the year. These measurements underline that beside higher temperature and higher vascular plant proportion, the top 20 cm peat at the High T site stays longer aerated over the year than at the Low T site. This information will be replacing the information of water table depths at the end of the sentence in L. 89 as follows:*

**"The time during which the top 20 cm of the peat was above the water table was determined with water table measurements between August 2015 and July 2016 at three gauges on each site. At the Low T site, the water table remained below 20 cm for 137, 138 and 284 days of the year; at the High T site, this was 117, 360 and 366 days of the year respectively**."

*Due to this added information, we will adapt the discussion in L. 308 as follows:*
"The altitudinal gradient has been used to reveal potential effects of increasing temperature **and associated lower water table** on peat decomposition by comparing the decomposition parameters (Table 1) between the High **T** and Low T site.

*Water table and thus aeration did probably not affect peat decomposition under sedges vs. shrubs because our sampling design ensured similar water table between sedges and shrubs at each of our sites. We think that increased peat decomposition as a result of oxygen transportation by the aerenchym of sedges should be more important at the Low T site because of the longer time of water saturation at this site in comparison to the high T site. We will address this in L. 305 as follows:*

**"This process is particularly relevant at the Low T site, where the uppermost 20 cm of the peat remained water saturated much longer than at the High T site."**

*In addition, we will refine our conclusions in L. 348:*
"The most important and also surprising result of our study was that vascular plants had a more pronounced impact on peat decomposition than temperature **and associated lower water tables together. Potential O$_2$ transport by the aerenchym of sedges did probably not contribute to enhanced peat decomposition at the High T site, as the top 20 cm peat layer sampled remained above the water table, and thus aerated, for most of the year.**"

- The authors tend to use general terms such as
L 35 alterations in the environment,
*We thank the referee for the comment. We will address this in L. 35 as follows:*
"Climate change is expected to partly lift these environmental constraints to microbial decomposition **by warmer** (Karhu et al., 2014) **and drier conditions**, threatening to release stored organic C as CO$_2$ to the atmosphere."

L39 plant-soil feedbacks,
*We will change the term in L. 39 as follows:*
"A systematic change in composition of plant functional types (PFTs) towards vascular plants has a yet unknown potential to accelerate C losses from the **stored** peat originally dominated by mosses **due to increased C input via roots from vascular plants** (Bragazza et al., 2013; Gavazov et al., 2018; Robroek et al., 2015), **and litter mixing effects (Zhang et al., 2019)."**

L66..hydrological conditions. Please be more specific.

*We will change the term in L. 66 as follows:*

"However, stable isotope patterns are also affected by **the water table limiting aerobic decomposition and thus isotopic discrimination in the remaining peat (Krüger et al., 2015) and the** plant species forming the litter."

- Can the authors give an estimate about the ages of their plant/peat samples. Is time an issue here?

*Unfortunately, we don't have data on the age of peat in 20 cm depth of both sites. A recent study of two other alpine peatlands at an altitudinal contrast higher than our study (1030 m a.s.l. vs. 1880 m a.s.l.) reported peat ages of 40 and 26 years respectively for the peat in 15-20 cm depth (Gavazov et al., 2018). Assuming these peatlands are similar to ours, the age difference between our sites would be less than 15 years.*

*The differences between the sites may have affected absolute differences between peat decomposition, but not the impact of shrubs relative to sedges on peat decomposition within sites. Furthermore, differences between the sites were quite small, and we are not able to disentangle the effects of the current temperature and water table on the one hand, and differences in environmental condition during peat formation on the other hand.*

*Accordingly, we will add information on the estimated age of peat at the end of the sentence in L. 90 in the study site descriptions as follows:*

"**Furthermore, in peatlands with similar vegetation cover, situated at 1030 m a.s.l. and 1880 m a.s.l. in Switzerland, the age of peat in 15-20 depth was found to be 40 years and 26 year, respectively (Gavazov et al., 2018), meaning that the potential age difference between the peat sampled at our sites is likely less than 15 years.**"

- L185-195 The description of Fig. 1 2 and 4 are a bit difficult to decipher. May be distinguish by site.

*As suggested, the paragraph in L.185-195 describing the results presented in these figures will be restructured as follows, but Figures 1, 2, and 4 will be maintained as we submitted them:*

"Carbon to N ratios and isotopic composition of plant shoots were different between **plant tissues,** PFTs and to a lesser extent between sites (Fig. 1). **Living plant shoots had consistently lower C/N ratios than plant roots irrespective of site or PFT (Fig. 1a). Sedge shoots had significantly lower C/N ratios (p < 0.001, Fig. 1a) and were significantly enriched in $^{13}$C (p < 0.001, Fig. 1b) compared to moss and shrub shoots. Shrub shoots were significantly depleted in $\delta^{15}$N compared to sedge and moss shoots (p < 0.001, Fig. 1c). Sedge shoots and moss from the High T site were both significantly depleted in $^{13}$C compared to Low T site (both p < 0.01, Fig. 1b).**

**Peat C/N ratio and isotopic composition was significantly affected by depth, site and to a lesser extent by PFTs (Fig. 2a, 2b). Carbon to N ratio increased with depth and was higher at the High T site compared to the Low T site (Fig. 2a, Table 2). In the upper 0-2 cm peat layer, C/N ratios and stable isotopes corresponded with values observed for moss (Fig. 1, Fig. 2). Isotopic composition of peat increments was significantly different for depths and PFTs (Table 2). $\delta^{13}$C and $\delta^{15}$N of peat increased with depth (Fig. 2b, 2c) and the enrichment in $^{13}$C with depth was stronger for sedge-cores than for shrub-cores (i.e. 2.7 ± 0.4 ‰ for sedge-cores and 1.6 ± 0.4 ‰ for shrub cores). The largest enrichment in $\delta^{13}$C between the uppermost and lowest depth increment appeared in sedge-cores at the High T site (Fig. 2b; i.e., 3.5 ± 0.5 ‰).**"

- Table 1: all reference are from one of the authors (Schellekens). Any independent references available here?

*The information on peat pyrolysates is frequently fragmented and scattered in the literature and the identification and/or validation of parameters was mainly demonstrated by these studies of Schellekens et al. However, we fully agree with the reviewer that self-citing is not desired and that the original source publication must be cited too. In order to cite correctly and at the same time avoid repeating the whole reasoning behind the pyrolytic parameters*

*(that was already published in Schellekens et al.), we have checked the literature once more and we will revise the references accordingly as follows (both in Table 1 and in Section 2.5): Also, note that the reference is given for the interpretation in Sphagnum peat (4th column) and not for the source molecule (3rd column); this was not sufficiently clear and perhaps contributed to the reviewers comment. We will clarify this in the revised manuscript by indicating that interpretation refers to Sphagnum peat; see the heading of the fourth column in Table 1.*

| Parameter | Unit | Indication | Interpretation in *Sphagnum*-dominated peat |
|---|---|---|---|
| C/N | - | preferential decomposition of C over N | aerobic decomposition |
| $\delta^{13}C$ | [‰] | preferential decomposition of $^{12}C$ over $^{13}C$ isotope | aerobic decomposition |
| $\delta^{15}N$ | [‰] | preferential decomposition of $^{14}N$ over $^{15}N$ isotope | aerobic decomposition |
| sum of G and S lignin products | [% TIC[a]] | lignin | vascular plants (van Smeerdijk and Boon, 1987) |
| sum of *n*-alkenes and *n*-alkanes | [% TIC[a]] | cutan, suberan, leaf waxes | ericoid shrubs (Schellekens and Buurman, 2011; van Smeerdijk and Boon, 1987) |
| $C_3G/G$ | - | intact lignin | ericoid shrubs (Schellekens et al., 2012) |
| 4-vG/G | - | ferulic acid | sedges (van Smeerdijk and Boon, 1987; Schellekens et al., 2012) |
| 4-isopropenylphenol (Ph6) | [% TIC[a]] | sphagnum acid | aerobic decomposition of *Sphagnum* tissues (preferential loss of polyphenols over polysaccharides; Schellekens et al., 2015b) |
| levoglucosan/sum of polysaccharides | - | cellulose | aerobic decomposition of *Sphagnum* tissues (preservation of *Sphagnum* polysaccharides; Schellekens et al., 2015b)[b] |

[a] total ion current
[b] interpretation refers to relatively low values

*We will update the paragraph in L. 141ff. as follows:*

"Based on the results of previous pyrolysis studies from peatlands a number of pyrolytic parameters reflecting plant species and the degree of **peat** decomposition were extracted (Table 1). A pyrolysis product specific for sphagnum acid (4-isopropenylphenol; Van Der Heijden et al., 1997) has been found to very sensitively reflect aerobic decomposition of *Sphagnum* tissue in *Sphagnum*-dominated peat **(Schellekens et al., 2015b)**. Methoxyphenols are unique to lignin, thereby providing a measure for the contribution from vascular plants in peat dominated by *Sphagnum*, because *Sphagnum* contains no lignin (Abbott et al., 2013; Kracht and Gleixner, 2000**; Schellekens et al., 2015c; van Smeerdijk and Boon, 1987).** Since both shrubs and sedges contain lignin, additional parameters were included to distinguish between them. Sedges have large contributions from p-coumaric and ferulic acid (Lu and Ralph, **1999**) with typical pyrolysis products 4-vinylphenol (Lg1) and 4-vinylguaiacol (Lg4), respectively (**van der Hage et al., 1993**). Because 4-vinylphenol is also abundant in *Sphagnum* tissue **(van Smeerdijk and Boon, 1987)**, the ratio of 4-vinylguaiacol to the summed guaiacyl products (G) **can** be used to reflect sedges (Schellekens et al., 2012). The ratio of $C_3$-guaiacol to G **usually reflects intact lignin in soils but has been found** indicative for shrubs **in peat** (Schellekens et al., 2012, 2015a). *n*-Alkenes and *n*-alkanes (Al) originate from cutan and suberan present in roots and bark (**Nierop, 1998; Tegelaar et al., 1995**) and leaf waxes (Eglinton and Hamilton, 1967), depending on their chain length, all of which are associated with shrubs in *Sphagnum*-dominated peat (Schellekens and Buurman, 2011**; van Smeerdijk and Boon, 1987**)."

*Furthermore, we have checked the whole text for possible reduction of self-citing. In L.73 we will exclude Schellekens et al., 2009 and 2015c.*

*Further, we will change typography and make other corrections:*

*Sentence in L. 77 will be changed to:*
"In this multi-proxy study, we combined the analytical approaches outlined above to explore the influence of vascular plants on chemical properties and degree of **peat** decomposition in **two** moss-dominated peatlands contrasting in temperature."

*We will correct the following references in our manuscript:*
- L. 147f where (Lu and Ralph, 2010) will be changed to **Lu and Ralph, 1999**
- L.278 where Schellekens et al., 2015a will be changed to **Schellekens et al., 2015b**
- L.317 where Schellekens et al., 2015c will be changed to **Schellekens et al., 2015b**

*Commas will be included in:*
- L. 324: "Given the above**,** …"
- 4.2.2 too**,**

*Changes to italic will be done in:*
- L. 336 *E. vaginatum*
- L. 327 *Sphagnum*-dominated

*Grammar in L. 243 will be corrected to:*
"Furthermore, the similarity of C/N ratios, $\delta^{13}C$ and $\delta^{15}N$ of the uppermost peat increment to those of moss are indicative for moss-dominated peat (Schaub and Alewell, 2009) and **has** been measured likewise in *Sphagnum* peatlands by Kracht and Gleixner (2000)."

*We will adjust the sentence in L. 247 as follows:*
"**Peat composition under shrubs and sedges is influenced by these species in the studied peat (0-20 cm) as indicated by the molecular parameters for sedge and shrub in the corresponding peat cores (Fig 4b, 4c, 4d).**"

*We will make changes in L. 274*:
"In addition to changes in $\delta^{13}C$ and $\delta^{15}N$ reflecting the decomposition of the bulk peat (i.e. cumulative effects on all peat components), we examined changes in compounds being indicative for the decomposition of specific plant tissues, i.e. *Sphagnum*-derived peat (**4-isopropenylphenol**)."

*We will make changes in L. 295:*
"**Because this is not evident from the 4-isopropenylphenol record, it probably reflects a** higher contribution of sedge-derived polysaccharides **at these depths**."

*The sentence in L. 296f. ("Such a shift….") will be deleted.*

*We will make changes in L. 298:*
"The observed **decomposition** patterns were detected by a parameter describing the whole peat ($\delta^{13}C$), and were also reflected by compounds indicative for *Sphagnum* material (4-isopropenylphenol)."

*The Figure reference in the sentence in L. 302 will be specified:*
"Sedge litter is likely to be more readily decomposable compared to shrub litter, caused by its lower C/N ratios (Fig. 1**a**; Huang et al., 1998; Kaštovská et al., 2018; Laiho et al., 2003; Limpens and Berendse, 2003).

*We will make changes in L. 336:*
"On the other hand, Zeh et al. (2019) could show that shrubs translocated more C into the peat at

[revised manuscript text omitted]

Hobbie, S. E. and Chapin, F. S.: Response of tundra plant biomass, aboveground production, nitrogen, and CO2 flux to experimental warming, Ecology, 79(5), 1526–1544, doi:10.1890/0012-9658(1998)079[1526:TROTPB]2.0.CO;2, 1998.

*As already addressed to Tim Moore, in L. 375 the Biester et al. reference occurs twice in the literature of which one will be deleted. Accordingly, Biester et al., 2014a will be changed to Biester et al., 2014.*

*As already addressed to Tim Moore, in L. 559 the Ward et al. reference will be corrected to:*

"Ward, S. E., Ostle, N. J., Oakley, S., Quirk, H., Henrys, P. A. and Bardgett, R. D.: Warming effects on greenhouse gas fluxes in peatlands are modulated by vegetation composition, Ecol. Lett., 16(10), 1285–1293, doi:10.1111/ele.12167, 2013."

---

## Author Comment (AC2) · 7 Apr 2020

We thank Tim Moore for the very helpful and constructive comments on our work, which helped us to further improve the manuscript. Therefore, we tried to consider all of them.

The referee's comments will we shown black. Our response is shown blue, *italic* and tab-indented while changes in the manuscript are in quotation marks and **bold**.

**Referee #2: Tim Moore**

This manuscript examines the influence of variations in plant cover on the rate of decomposition in the upper layers of the peat profile, at two sites in the Italian Alps, which vary in their mean annual temperature. The aim is to provide some indication of what may happen if climate change warms peatlands and the vegetation cover of sedges (here Eriophorum) and shrubs (here Calluna) increases at the expense of Sphagnum moss. Peat cores were analyzed for a wide range of properties, related to degree of decomposition, including pyrolysis, which is unusual in peat studies. The results suggested that both temperature (over a 1.4oC range) and vegetation cover influenced decomposition rate of the peat, dominated by residual moss, but that changes in vegetation to sedges and to a lesser shrubs, were more important than the temperature rise, using the two sites as proxies for change.

This contribution is one of several suggesting that changes in vegetation from global change are likely to be more important than simple rises in temperature in affecting the C budget of ecosystems, for example the 'shrubification' of the Arctic. Here, detailed and careful sampling of two sites, with modest differences in air temperature but varying in vascular plant coverage 47 and 77%), provide a suitable analogue to address this issue. The peat samples, and vegetation, have been analyzed by a variety of techniques, some of them common, such as elemental and stable isotopes, and some less common, such as gc/ms pyrolysis. The content of the manuscript is suitable for Biogeosciences and it is generally well written, though I have noted a few errors of the pdf, along with some specific comments.

Some comments for 'discussion':
The sites vary in terms of their mean annual temperature, but is this translated to similar differences in the peat layer undergoing decomposition? Are soil temperature data available to be more precise on the thermal differences in the peat at the two sites? It might be smaller or larger than the 1.4oC.

*We thank Tim Moore for this comment. We will give more detailed information on temperature differences in 10 cm depth at the end of the sentence in L. 88 as follows:*
**"Soil temperature in 10 cm depth between August 2015 and July 2016 was 7.1°C at Lupicino and 5.9°C at Palù Tremole."**

Is there an estimate at the rate of peat accretion at the sites? In other words can you estimate over what period the 20 cm of peat have accumulated (e.g. by 210Pb dating, perhaps a century?) and what are the changes in the environment over that period? Is what we see now, the same as what it was a century ago, when the current 20 cm peat began to form? For line 263, can you provide an estimate of 'increasing time of exposure'?

*As already addressed to referee #1, we don't have data on the age of peat in 20 cm depth of both sites. A recent study of two other alpine peatlands at an altitudinal contrast higher than our study (1030 m a.s.l. vs. 1880 m a.s.l.) reported peat ages of 40 and 26 years respectively for the peat in 15-20 cm depth (Gavazov et al., 2018). Assuming these peatlands are similar to ours, the age difference between our sites would be less than 15 years.*
*The differences between the sites may have affected absolute differences between peat decomposition, but not the impact of shrubs relative to sedges on peat decomposition within sites. Furthermore, differences between the sites were quite small, and we are not able to*

*disentangle the effects of the current temperature and water table on the one hand, and differences in environmental condition during peat formation on the other hand. Accordingly, we will add information on the estimated age of peat at the end of the sentence in L. 90 in the study site descriptions as follows:*

"**Furthermore, in peatlands with similar vegetation cover, situated at 1030 m a.s.l. and 1880 m a.s.l. in Switzerland, the age of peat in 15-20 depth was found to be 40 years and 26 year, respectively (Gavazov et al., 2018), meaning that the potential age difference between the peat sampled at our sites is likely less than 15 years.**"

Do you have any estimates of the input of litter into the sites, based on the vegetation composition, to provide a quantitative context of 'how much' is being added? The references cited (lines 41-42) tended to be for Arctic tundra, which is presumably inapplicable to alpine conditions.

*We thank the referee for this comment and will specify the information in the sentence in L. 41 based on an applicable reference as follows:*

"Vascular plants **in alpine peatlands were shown to have up to twice as high net biomass production as mosses (Gerdol et al., 2010) and** consequently relatively higher litter inputs."

I think that careful attention should be given to the water table at the two sites which are reported on line 89. Perhaps the Zeh et al. (2019) ms contains more specific information, but a difference in water table of 30 cm (the minimums reported) would have a profound effect on decomposition rates in the peat cores, the High T site being both warmer and drier… Was August 2015 to July 2016 'typical' in terms of hydrology (i.e. precipitation etc.)? On the other hand, if the highest water table measured was 17 and 15 cm, it means that only the bottom 3 to 5 cm of the 20 cm core were at and under the water table, so we are dealing with decomposition under aerobic conditions, effectively the acrotelm. Perhaps a useful metric would be the proportion of the year in which the water table was within the 20 cm core, especially the 15-20 cm section, to see whether hydrology was significantly different at the two sites. An increased vascular cover, associated with a warming, will likely increase evapotranspiration rates, which in turn will produce a lowered water table, accelerating the vascular 'invasion'.

*We thank Tim Moore for his valuable comment on this topic. As already addressed to referee #1, sampling depths in the peat cores are periodically water saturated. Water table measurements 2015/08–2016/07 (three gauges at each site) show that the peat within the sampled 20 cm on the High T site stayed aerated for 117, 360 or 366 days of the year as recorded in the three gauges. On the Low T site, the peat of the top 20 cm was aerated for 137, 138 or 284 days of the year. These measurements underline that beside higher temperature and higher vascular plant proportion, the top 20 cm peat at the High T site stays longer aerated over the year than at the Low T site. This information will be replacing the information of water table depths at the end of the sentence in L. 89 as follows:*

"**The time during which the top 20 cm of the peat was above the water table was determined with water table measurements between August 2015 and July 2016 at three gauges on each site. At the Low T site, the water table remained below 20 cm for 137, 138 and 284 days of the year; at the High T site, this was 117, 360 and 366 days of the year respectively**."

*Due to this added information, we will adapt the discussion in L. 308 as follows:*

"The altitudinal gradient has been used to reveal potential effects of increasing temperature **and associated lower water table** on peat decomposition by comparing the decomposition parameters (Table 1) between the High **T** and Low T site."

Eriophorum is arenchymous, with the capacity to oxygenate the peat: would that influence the peat environment in terms of decomposition rate, given that the top 20 cm is above the water table for most of the year?

*As already addressed to referee #1, water table and thus aeration did probably not affect differences in peat decomposition under sedges vs. shrubs because our sampling design*

*ensured similar water table between sedges and shrubs at each of our sites. Water table controlled aeration is a difference between sites though differences in decomposition between PFTs are more pronounced.*

*We will address this in L. 305 as follows:*

**"This process is particularly relevant at the Low T site, where the uppermost 20 cm of the peat is water saturated much longer than at the High T site.**

*And further we will be more precise in our conclusions in L. 348:*

"The most important and also surprising result of our study was that vascular plants had a more pronounced impact on peat decomposition than temperature **and associated lower water tables together. Potential $O_2$ transport by the aerenchym of sedges did probably not contribute to enhanced peat decomposition at the High T site, as the top 20 cm peat layer sampled remained above the water table, and thus aerated, for most of the year."**

Line 190: I was surprised to see the large increase in C:N ratio with depth in nearly all the cores, whereas with decomposition one might have expected a decline in the ratio. Is there an explanation for this pattern – I could not see one in the Discussion (cf Table 1). Does atmospheric N deposition play a role here (larger N concentrations in the past few decades)?

*We thank Tim Moore for his suggestion and will add this information to the discussion section 4.2 as follows in L. 271:*

"Differences in the C/N ratios did not provide **a** consistent picture regarding changes in peat decomposition**. We speculate that the observed increasing C/N ratio with depth might reflect an increasing N deposition in the past decades (Galloway et al., 2008) and an** increased contribution of roots **(high C/N ratios, Fig. 1a) to** peat formation with increasing depth (Sect. 4.1.2). **N deposition at the High T site were reported to be 8.2 kg ha$^{-1}$ y$^{-1}$ (Bragazza et al., 2005) and for peatlands in norther Italy between 4.2 and > 10 kg ha$^{-1}$ y$^{-1}$ (Bragazza et al., 2003, 2005)."**

Line 230 I would think that there are major differences in 15N among the three plant types from zero to -10, which relate back to, I assume, the mycorrhizal dependance of Calluna, the non-mychorrizal Eriophorum and moss in between

*We thank Tim Moore for his suggestion and will include this into the discussion section in L. 226 as follows:*

"Properties as C/N ratios, $\delta^{13}$C and $\delta^{15}$N **of living plant parts** showed **significant** differences between different **PFTs** (Fig. 1). C/N ratios of sedge shoots were significantly lower than those of shrubs or mosses**, in line with its higher decomposability (Kaštovská et al., 2018; Limpens and Berendse, 2003)**. $^{13}$**C signature of plant tissue tends to be depleted under** warmer and drier conditions (Broder et al., 2012; Gavazov et al., 2016; Skrzypek et al., 2007) explaining the observed $\delta^{13}$C depletion of sedge and moss tissues at the High T site compared to Low T site. **Shrub shoots were significantly depleted in $\delta^{15}$N compared to sedge and moss shoots (p < 0.001, Fig. 1c). The depletion of $\delta^{15}$N in living *C. vulgaris* shoots compared to shoots of *E. vaginatum* and *Sphagnum spp.* is likely related to the symbiotic relationship between *C. vulgaris* and ericoid mycorrhizal fungi (Aerts et al., 2009; Bragazza et al., 2010; Emmerton et al., 2001). The transferred N from mycorrhiza to shrubs occurs to be depleted relative to soil N (Gebauer and Dietrich, 1993). Nevertheless, though differences in $\delta^{13}$C and $\delta^{15}$N between PFT existed, unknown variables influencing $\delta^{13}$C and $\delta^{15}$N in peat, such as various plant species growing in the peat, N deposition history etc. complicated the use of $\delta^{13}$C and $\delta^{15}$N as vegetation proxy."**

*As addressed to referee #1, the paragraph in L.185-195 will be restructured as follows, but Figures 1, 2, and 4 will be maintained as we submitted them:*

"Carbon to N ratios and isotopic composition of plant shoots were different between **plant tissues**, PFTs and to a lesser extent between sites (Fig. 1). **Living plant shoots had consistently lower C/N ratios than plant roots irrespective of site or PFT (Fig. 1a). Sedge shoots had significantly lower C/N ratios (p < 0.001, Fig. 1a) and were significantly enriched in $^{13}$C (p < 0.001, Fig. 1b) compared to**

**moss and shrub shoots. Shrub shoots were significantly depleted in δ15N compared to sedge and moss shoots (p < 0.001, Fig. 1c). Sedge shoots and moss from the High T site were both significantly depleted in 13C compared to Low T site (both p < 0.01, Fig. 1b).**
**Peat C/N ratio and isotopic composition was significantly affected by depth, site and to a lesser extent by PFTs (Fig. 2a, 2b). Carbon to N ratio increased with depth and was higher at the High T site compared to the Low T site (Fig. 2a, Table 2). In the upper 0-2 cm peat layer, C/N ratios and stable isotopes corresponded with values observed for moss (Fig. 1, Fig. 2). Isotopic composition of peat increments was significantly different for depths and PFTs (Table 2). δ13C and δ15N of peat increased with depth (Fig. 2b, 2c) and the enrichment in 13C with depth was stronger for sedge-cores than for shrub-cores (i.e. 2.7 ± 0.4 ‰ for sedge-cores and 1.6 ± 0.4 ‰ for shrub cores). The largest enrichment in δ13C between the uppermost and lowest depth increment appeared in sedge-cores at the High T site (Fig. 2b; i.e., 3.5 ± 0.5 ‰)."**

I got goggle-eyed looking at the symbols in Figures 1, 2 and 4 and would appreciate some differentiation stronger than washed-out blue and a yukky looking green. Be 'artistic'! Simple black and red would be nice…

*We thank the referee for his opinion on the color map. Colors and their effect are indeed very personal. Therefore, we want to keep the colors we used.*

4.2.2 is a 45 line 'paragraph' and it might be easier to digest if it was broken down into three paragraphs, each dealing with a specific theme. It is a 'confounding' system with multiple interpretations of results and the strength of the ms is the range of analyses conducted.

*We will subset the section into four paragraphs and focus the argumentation as Time Moore suggested:*

"The altitudinal gradient has been used to reveal potential effects of increasing temperature and **associated lower water table** on peat decomposition by comparing the suggested decomposition parameters (Table 1) between the High **T** and Low T site. Increases in δ13C with depth were higher at the High T site than at the Low T site (Fig. 2). Therefore, decomposition of the moss-dominated peat is likely to be increased at the High T site compared to the Low T site independent of the vascular plant species.

Depth trends of the *Sphagnum*-specific decomposition parameter (4-isopropenylphenol) do not **reflect**, however**, this increased peat decomposition at higher temperatures**. 4-**I**sopropenylphenol decreased **less** at the **High** T site compared to the **Low** T site (Fig. 4e)**. We might speculate a difference in temperature sensitivity between decomposition of Sphagnum and the whole peat, but such a hypothesis needs to be tested by e.g. temperature controlled incubation experiments. The ratio of levoglucosan to the sum of polysaccharides (Fig. 4f) did not show consistent trends related to temperature again indicating no temperature enhanced degradation of the moss-dominated peat. V**ascular plants and particularly sedges might increasingly contribute to polysaccharides with higher temperatures (see section 4.2.1). These changes from more Sphagnum-derived polysaccharides to more sedge-derived polysaccharides could change decomposition dynamics of polysaccharides, since cell walls constituents from Sphagnum were found to be less easily decomposable (Hájek et al., 2011). A higher contribution from sedges could therefore superimpose a potential enrichment of *Sphagnum*-derived polysaccharides with depth. Given the above, the higher degree of peat decomposition at the High T site picked up by δ13C is probably mediated by higher input rates of easily decomposable vascular plant litter, notably sedges, combined with the warmer and drier conditions favouring aerobic decomposition processes (Biester et al., 2014).

This combined effect of sedges and temperature on peat decomposition has implications for the long-term C storage in *Sphagnum*-dominated peatlands because of projected shifts from sedges to shrubs with climate change (Breeuwer et al., 2009). That change towards less sedges may partly offset temperature driven decomposition processes because of the observed enhancing effect of sedges on peat decomposition. Less sedges (i.e. more shrubs) should result in less peat degradation because (i) the decomposability of available litter (higher contribution of shrub litter) is reduced as its

chemical composition indicates less decomposability (Kristensen and McCarty, 1999; Ward et al., 2015), (ii) an increasing presence of shrubs (*C. vulgaris*) supresses belowground biota activity and nutrient cycling (Fenner and Freeman, 2011), (iii) *C. vulgaris* associates with mycorrhizal fungi which increase the uptake of organic nutrients leading to an increase of C/N ratio (Read et al., 2004) and thus a decrease in peat decomposition (Ward et al., 2015), (iv) the input of labile C into peat via sedge roots is lower (Crow and Wieder, 2005; Robroek et al., 2015), (v) the transport of oxygen into peat via aerenchym of *E. vaginatum* is lower. On the other hand, Zeh et al. (2019) could show that shrubs translocated more C into the peat at higher temperatures than sedges, which should result in reinforcing effect on peat decomposition with increasing temperature. Obviously, the enhanced C input by shrubs into peat did not coincide with enhanced decomposition of the moss-dominated peat on these sites."

In the Conclusion, or somewhere in the Discussion, it would be useful to identify the 'bang for the buck' in these analyses: some are simple and routine and some, especially the gc/ms pyrolysis, is 'labour intensive'. Do you have anything to add to the Biester et al. 2014 paper, based on this specific application?

> *We thank Tim Moore for his comment. The conclusion from Biester et al. (2014) that py-GC/MS is particularly useful in disentangling effects of changes in vegetation composition and decomposition upon changes in environmental conditions (i.e. the hen and egg problem outlined by reviewer 1) is our starting point. In our section "conclusions", we addressed the advantages/disadvantages of the applied methods (L.343 – 348). As expected (also from Biester et al., 2014), the combination of different methods will result in a more comprehensive picture about peat decomposition. Without py-GC/MS we would not be able to conclude that the moss-dominated peat was more decomposed under sedges than shrubs (i.e. by applying 4-isopropenylphenol to reflect aerobic decomposition of Sphagnum tissue). At the time of the Biester paper (2014), it was not yet known that 4-isopropenylphenol reflects aerobic decomposition of Sphagnum tissue in Sphagnum-dominated peat (Schellekens et al., 2015b).*
> *The central objective of Biester et al 2014 is to compare analytical methods to determine peat decomposition, while our objective is to evaluate the effect of PFT on peat decomposition. We feel that these different themes are clear from the titles, and throughout our manuscript. Therefore, and considering the changes to Section 4.2.2, we do not see the necessity to further enlarge this part of the conclusions.*

I provide a copy of the pdf which I have annotated with comments and suggested typographical and other correction. https://www.biogeosciences-discuss.net/bg-2019-503/bg-2019-503-RC2-supplement.pdf

Tim Moore

*Changes in typography and other suggestions from Tim Moore's pdf*

L. 42 "Vascular plants have a higher biomass production (Hobbie and Chapin, 1998) and consequently relatively higher litter inputs than mosses (Elmendorf et al., 2012)." These (Elmersdorf) are results based on work in arctic tundra, whereas your site is alpine, so it is perhaps not very applicable.

> *As already stated to referee #1, we will specify the information in the sentence in L. 41 based on an applicable reference as follows:*
"Vascular plants **in alpine peatlands were shown to have up to twice as high net biomass production as mosses (Gerdol et al., 2010)** and consequently relatively higher litter inputs."

L. 70 and range of 15N in Sphagnum?

> *We will add information about $^{15}$N rages in Sphagnum at the end of the sentence in L. 70 as follows:*

„Sedge leaves were found to be more enriched in $^{13}$C and $^{15}$N than shrub leaves. **The isotopic ratios for living plant parts found in this study are consistent with the ranges reported in previous studies. Sedge leaves were found to vary in δ$^{13}$C signature between -27.0 to -26.85 ‰ and in δ$^{15}$N between -3.96 to -0.9 ‰; reported ranges for δ$^{13}$C in shrub leaves are -29.2 to -28.83 ‰ and for δ$^{15}$N -10.92 to -9.7 ‰ (Biester et al., 2014; Gavazov et al., 2016; Ménot and Burns, 2001; Nordbakken et al., 2003).** *Sphagnum* samples **were found to** vary in δ$^{13}$C signature between **-30.4 and -25.0** ‰ (Bragazza and Iacumin, 2009; **Loisel et al., 2009;** Preis et al., 2018; Price et al., 1997; Proctor et al., 1992**) and in δ$^{15}$N signatures between -7.5 and 2.5 ‰ (Asada et al., 2005; Biester et al., 2014; Bragazza et al., 2005; Kohzu et al., 2003; Ménot & Burns, 2001; Nordbakken et al., 2003).**"

*L. 50 The word "microorganism" will be corrected to the plural*:
"**microorganisms**"

*In L. 375 the Biester et al. reference occurs twice in the literature of which one will be deleted. Accordingly, Biester et al., 2014a will be changed to:*
**Biester et al., 2014**

*In L. 559 the Ward et al. reference will be corrected to:*
"**Ward, S. E., Ostle, N. J., Oakley, S., Quirk, H., Henrys, P. A. and Bardgett, R. D.: Warming effects on greenhouse gas fluxes in peatlands are modulated by vegetation composition, Ecol. Lett., 16(10), 1285–1293, doi:10.1111/ele.12167, 2013.**"

*Further, we will change typography and make other corrections:*

*Sentence in L. 77 will be changed to:*
"In this multi-proxy study, we combined the analytical approaches outlined above to explore the influence of vascular plants on chemical properties and degree of **peat** decomposition in **two** moss-dominated peatlands contrasting in temperature."

*As already addressed to referee #1, we will update the paragraph in L. 141ff. as follows:*
"Based on the results of previous pyrolysis studies from peatlands a number of pyrolytic parameters reflecting plant species and the degree of **peat** decomposition were extracted (Table 1). A pyrolysis product specific for sphagnum acid (4-isopropenylphenol; Van Der Heijden et al., 1997) has been found to very sensitively reflect aerobic decomposition of *Sphagnum* tissue in *Sphagnum*-dominated peat **(Schellekens et al., 2015b**). Methoxyphenols are unique to lignin, thereby providing a measure for the contribution from vascular plants in peat dominated by *Sphagnum*, because *Sphagnum* contains no lignin (Abbott et al., 2013; Kracht and Gleixner, 2000**; Schellekens et al., 2015c; van Smeerdijk and Boon, 1987).** Since both shrubs and sedges contain lignin, additional parameters were included to distinguish between them. Sedges have large contributions from p-coumaric and ferulic acid (Lu and Ralph, **1999**) with typical pyrolysis products 4-vinylphenol (Lg1) and 4-vinylguaiacol (Lg4), respectively (**van der Hage et al., 1993**). Because 4-vinylphenol is also abundant in *Sphagnum* tissue **(van Smeerdijk and Boon, 1987)**, the ratio of 4-vinylguaiacol to the summed guaiacyl products (G) **can** be used to reflect sedges (Schellekens et al., 2012). The ratio of $C_3$-guaiacol to G **usually reflects intact lignin in soils but has been found** indicative for shrubs **in peat** (Schellekens et al., 2012, 2015a). *n*-Alkenes and *n*-alkanes (Al) originate from cutan and suberan present in roots and bark (**Nierop, 1998; Tegelaar et al., 1995**) and leaf waxes (Eglinton and Hamilton, 1967), depending on their chain length, all of which are associated with shrubs in *Sphagnum*-dominated peat (Schellekens and Buurman, 2011**; van Smeerdijk and Boon, 1987**)."

*We will correct the following references in our manuscript:*
- L. 147f where (Lu and Ralph, 2010) will be changed to **Lu and Ralph, 1999**
- L.278 where Schellekens et al., 2015a will be changed to **Schellekens et al., 2015b**
- L.317 where Schellekens et al., 2015c will be changed to **Schellekens et al., 2015b**

*Commas will be included in:*
- L. 324: "Given the above**,** …"
- 4.2.2 too**,**

*Changes to italic will be done in:*
- L. 336 *E. vaginatum*
- L. 327 *Sphagnum*-dominated

*Grammar in L. 243 will be corrected to:*
"Furthermore, the similarity of C/N ratios, $\delta^{13}$C and $\delta^{15}$N of the uppermost peat increment to those of moss are indicative for moss-dominated peat (Schaub and Alewell, 2009) and **has** been measured likewise in *Sphagnum* peatlands by Kracht and Gleixner (2000)."

*We will adjust the sentence in L. 247 as follows:*
"**Peat composition under shrubs and sedges is influenced by these species in the studied peat (0-20 cm) as indicated by the molecular parameters for sedge and shrub in the corresponding peat cores (Fig 4b, 4c, 4d).**"

*We will make changes in L. 274*:
"In addition to changes in $\delta^{13}$C and $\delta^{15}$N reflecting the decomposition of the bulk peat (i.e.

cumulative effects on all peat components), we examined changes in compounds being indicative for the decomposition of specific plant tissues, i.e. *Sphagnum*-derived peat (**4-isopropenylphenol**)."

*We will make changes in L. 295:*
"**Because this is not evident from the 4-isopropenylphenol record, it probably reflects a** higher contribution of sedge-derived polysaccharides **at these depths**."

*The sentence in L. 296f. ("Such a shift….") will be deleted.*

*We will make changes in L. 298:*
"The observed **decomposition** patterns were detected by a parameter describing the whole peat ($\delta^{13}C$), and were also reflected by compounds indicative for *Sphagnum* material (4-isopropenylphenol)."

*The Figure reference in the sentence in L. 302 will be specified:*
"Sedge litter is likely to be more readily decomposable compared to shrub litter, caused by its lower C/N ratios (Fig. 1**a**; Huang et al., 1998; Kaštovská et al., 2018; Laiho et al., 2003; Limpens and Berendse, 2003).

*We will make changes in L. 336:*
"On the other hand, Zeh et al. (2019) could show that shrubs translocated more C into the peat at higher temperatures than sedges, which **could** result in reinforcing effect on peat decomposition with increasing temperature."

*We will add these references to our literature:*

[revised manuscript text omitted]

---

## Author Response (AR3)

Dear Authors,

I went through your new manuscript and response to referee's comments. Your modifications improved much your manuscript, but I detect some important issues in the introduction section. This section is completely desorganized with a first paragraph made of 33 lines! A paragraph shoud ideally comprise between 10 to 15 lines. The objectives of paragraphs are not clearly separated. For example, the last paragraph that should insert the objectives of the study starts with the presentation of the py-GC/MS method. Another example is the definition of peat that arrives in the middle of your introduction though the subject is discussed from the first line. The writting of your introduction must be structured in paragraphs with clearly-identified messages.

Regards,

Sébastien

Dear Sebastien Fontaine,

we agree, that the introduction section looks disorganized and this is due to formatting. In two cases a paragraph ends exactly with the right margin. We now emphasized the paragraphs with blank lines, but in the final format, they should not appear. Without intention, one paragraph indeed was deleted and we now included it.

We intensively discussed the most appropriate position in the introduction to define peat. We expect that the reader of Biogeosciences has an idea about peat. We think that the best position in the text would be before describing in detail the properties of peat and potential relationships to decomposition. Therefore, we would prefer to not change the position in the manuscript where peat was defined.

Best Regards,

Lilli Zeh

---

## Author Response (AR4)

Hello,

I read your introduction and found your structure clearer. Nevertheless, the place of the peat definition is not relevant. It should appear either at the beginning of the introduction or the material & methods section. The definition of the peat decomposition could also appear in the material & methods section.

L67: The stable isotopes are often used for what purpose? Need to be clarified.

Regards,

Sébastien

Dear Sébastien Fontaine,

thank you for your comment on the place of peat- and decomposition definition. We fully agree that these definitions fit quite well into the material and methods section. As suggested, we now integrated these definitions in section 2.4 in l. 140 - 143.

We further clarified the use of stable isotopes in I. 67 f.

For overview, changes are tracked and attached to this letter.

Best regards,

Lilli Zeh

**Vascular plants affect properties and decomposition of mossdominated peat, particularly at elevated temperatures**

Lilli Zeh1, Marie Theresa Igel1, Judith Schellekens2, Juul Limpens3, Luca Bragazza4, Karsten Kalbitz1

[revised manuscript text omitted]

---

## Author Response (AR5)

Hello,

the transfer of these définitions in the material & methods section are welcome. However, the modified sentence above these définitions has ne more sense. Could you please ask to supervisors or co-author senior scientists to re-read your manuscript?

Regards

Sébastien

Dear Dr. Fontaine,

We re-discussed the most appropriate position to define peat and changed it accordingly as identified by track changes in the pdf file (section 2.2., fourth sentence, "Peat was defined as all organic bulk material accumulating underneath the peatlands surface, comprised of a matrix of mostly dead *Sphagnum* material with embedded living and dead stems and roots of *C. vulgaris* and *E. vaginatum*"). We also adapted the sentence before we define peat decomposition (section 2.4. first sentence: "To identify chemical properties of PFTs, peat and peat decomposition, representative plant samples for each PFT and one shrub-core and one sedge-core from each peatland were selected to be additionally analysed by py-GC/MS"). Further, we carefully re-read the manuscript and did some minor corrections. We hope that our manuscript is now suitable for publication.

Best Regards,
Lilli Zeh

[revised manuscript text omitted]